# Learning State-Based Node Representations from a Class Hierarchy for Fine-Grained Open-Set Detection

**Spandan Pyakurel** [1]   **Qi Yu** [1]

## Abstract

Fine-Grained Openset Detection (FGOD) poses a fundamental challenge due to the similarity between the openset classes and those closed-set ones. Since real-world objects/entities tend to form a hierarchical structure, the fine-grained relationship among the closed-set classes as captured by the hierarchy could potentially improve the FGOD performance. Intuitively, the hierarchical dependency among different classes allows the model to recognize their subtle differences, which in turn makes it better at differentiating similar open-set classes even they may share the same parent. However, simply performing openset detection in a top-down fashion by building a local detector for each node may result in a poor detection performance. Our theoretical analysis also reveals that maximizing the probability of the path leading to the ground-truth leaf node also results in a sub-optimal training process. To systematically address this issue, we propose to formulate a novel state-based node representation, which constructs a state space based upon the entire hierarchical structure. We prove that the state-based representation guarantees to maximize the probability on the path leading to the ground-truth leaf node. Extensive experiments on multiple real-world hierarchical datasets clearly demonstrate the superior performance of the proposed method.

## 1. Introduction

Openset detection aims to identify data samples that are not part of the known set (*a.k.a.*, closed-set) of classes in the training data. Most existing methods try to assign the detected openset samples into a single openset class (Sun et al., 2020; Bendale & Boult, 2016; Vaze et al., 2021; Chen

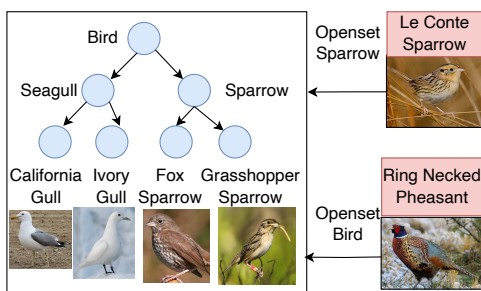

*Figure 1.* With the support of a hierarchy, fine-grained openset detection can assign an openset data sample into different parent nodes as shown: Le Conte Sparrow → Openset Sparrow, Ring Necked Pheasant → Openset Bird

et al., 2020b; 2021; Yang et al., 2020; Zhang & Patel, 2016). They typically quantify an openset score to measure the difference between the openset sample and all the closed-set classes as a whole. However, the semantic similarity between openset samples with closed-set classes may vary significantly. As the similarity increases, openset detection becomes more challenging and such problems are referred to as near OOD detection (Fang et al., 2022; Ren et al., 2021; Fort et al., 2021). Theoretical results have also been developed to strictly quantify the difficulty of these problems from the learnability perspective (Fang et al., 2022).

In many real-world settings, the fine-grained relationship between the closed-set classes can be precisely captured through a hierarchical structure. Such a hierarchy could help to improve the detection performance of some openset examples despite having a high semantic similarity with certain closed-set classes. As shown in Figure 1, the closed-set training data consists of four types of birds, which can be further grouped into two parent classes, Seagull and Sparrow. Due to migration of birds from other regions, one may need to identify previously unseen species. Identifying the closest related species of these new birds can help expand the existing hierarchy and provide insights into migration patterns. By training a model that adequately captures the hierarchical dependencies among different closed-set classes, it can be better equipped to more accurately recognize openset samples with different levels of similarity with the closed-set ones. For example, when being presented with a Le Conte Sparrow, the knowledge learned from the hierarchy could

[1]Rochester Institute of Technology, Rochester, New York. Correspondence to: Qi Yu <qi.yu@rit.edu>.

*Proceedings of the 42$^{st}$ International Conference on Machine Learning*, Vancouver, Canada. PMLR 267, 2025. Copyright 2025 by the author(s).

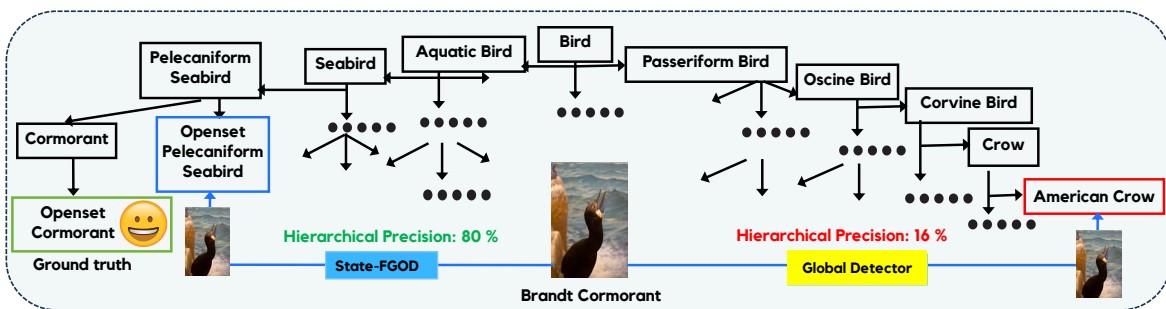

*Figure 2.* Error prediction analysis for a global detector that assigns an openset sample from Brandt Cormorant to a totally wrong class due to lack of considering the hierarchical dependencies.

allow the model to recognize some important but subtle differences from the two known Seagull classes. Meanwhile, it is also more different from the two Seagull classes. As a result, the model is able to perform *fine-grained openset detection (FGOD)* to recognize Le Conte Sparrow as an openset sparrow as shown in Figure 1. Similarly, when facing a Ring Necked Pheasant, the model could recognize that it is different from both Seagull and Sparrow while possessing some key characteristics of birds, and hence assigns it as an openset bird.

As illustrated by the example above, detecting openset samples that share semantic similarity with the closed-set classes (*i.e.,* near OOD detection) can be achieved by conducting FGOD when the closed-set classes can be organized into a hierarchical structure. Such hierarchical structures commonly exist in real-world settings as evidenced by many benchmark datasets (*e.g.,* Imagenet and Cifar100). To achieve fine-grained detection, an openset sample should be assigned to the right parent node in the hierarchy based on its difference to classes at different levels. Following the hierarchy, FGOD can be conveniently conducted in a top-down fashion by building a local detector at each non-leaf node. Each detector performs the classification of child nodes under the non-leaf node while quantifying an openset score. If a detector obtains a high openset score, an openset sample is detected. Otherwise, one child node is selected based on the classification result and the process continues unless a leaf node is reached.

However, using a sequence of local detectors in a top-down fashion may result in a poor FGOD performance. Each detector is trained by only considering its own child node distribution, which fails to fully utilize fine-grained dependencies from nodes at lower levels or those located at other parts of the hierarchy. This usually leads to local detection decisions that are sub-optimal. To train more robust detectors, one can collectively consider all the ancestor nodes on the path to the ground-truth leaf node. By choosing the path with the highest overall probability, the model can more effectively avoid poor local decisions. Nevertheless, a deeper analysis reveals that solely maximizing the (ground-truth) path probability does not guarantee to decrease the probability of other nodes that are not the children of the nodes on the path, which compromises the detection performance, as evidenced by our experiments.

To avoid large error accumulation of local detectors, a global detector can be trained that maximizes the probability of the ground truth node by considering all the leaf nodes. For open-set detection, besides evaluating an open-set score, an extra child node can be added to each non-leaf node denoting its open-set child class (Lee et al., 2018; Pyakurel & Yu, 2024), which transforms open-set detection as a multi-class classification problem that only considers all leaf nodes (see Figure 8 in the Appendix for an illustrative example). However, a global detector trained in such a way ignores the detailed hierarchical dependencies among different classes. Our thorough theoretical analysis reveals that the global detector formulation indeed leads to a sub-optimal training process causing large hierarchical detection errors. Such errors can be more precisely captured by the hierarchical performance metrics that measure the number of correct predictions out of all predictions along the hierarchy to reach the leaf node as shown in Figure 2.

To address the fundamental limitations of both local and global detectors for FGOD, we propose to learn novel state-based node representations from a hierarchy of closed-set classes to advance the frontiers of fine-grained open-set detection. Through the formulation of *legal states*, we can avoid error accumulation like in local detectors while effectively capturing fine-grained hierarchical dependencies between different nodes, including (parent → child), (ancestor → descendant), and (sibling → sibling). Such a formulation ensures that the probabilities assigned to each node (both leaf and non-leaf) strictly follow their hierarchical dependencies with theoretical guarantees. This implies that node probabilities are derived from a global view of the entire hierarchy. It is worth noting that there are a few existing works, such as (Deng et al., 2014; Chen et al., 2022), that perform state formulation, but they are designed for hierarchical classification instead of fine-grained openset

detection. Our novel theoretical analysis shows that the use of state-based representations in model training guarantees to decrease the probabilities of all non-ground truth nodes while increasing the ground-truth node's probability. Since these non-ground truth nodes include both leaf and non-leaf nodes, the prediction of the proposed method, even when wrong is close to the ground truth node in the hierarchy as shown in Figure 2, resulting in a higher hierarchical precision. Our contribution is threefold:

- a thorough theoretical analysis that unveals fundamental limitations of local and global detectors when performing open-set detection in a hierarchical structure,
- formulation of novel state-based node representations from a hierarchy of closed-set classes along with a FGOD process that avoids error accumulation in open-set detection while effectively capturing fine-grained hierarchical dependencies between different nodes,
- a formal analysis of the proposed state-FGOD model that ensures an optimal training process and guarantees to improve the hierarchical detection performance.

Extensive experimentation conducted on multiple real-world hierarchical datasets demonstrates the superior performance of the proposed method.

## 2. Related Works

**Openset detection.** Openset detection usually rely on a score to differentiate openset samples from closed-set classes. Openmax (Bendale & Boult, 2016) redistributes the closed-set probabilities and assigns a probability as a score to the openset class. Some works utilize the maximum softmax probability from the classification of closed-set classes for detection (Hendrycks & Gimpel, 2016; Vaze et al., 2021). Vaze et al. (Vaze et al., 2021) replace the probability with logit and utilize the maximum logit score for openset detection. Some works (Chen et al., 2020b; 2021; Yang et al., 2020) focus on training prototypes of closed-set classes in the embedding layer and learning a representation of openset class far from all the prototypes. Based on the distance of openset sample from the closed-set prototypes, detection is performed. A line of research quantifies uncertainty measure to identify openset samples based on evidential uncertainty (Sensoy et al., 2018; Malinin & Gales, 2018; Charpentier et al., 2020), free energy (Liu et al., 2020; Wang et al., 2021; Du et al., 2021). Few works (Fang et al., 2022; Ren et al., 2021; Fort et al., 2021; Lang et al., 2024; Sun et al., 2023; Fu et al., 2023) particularly focus on near-OOD detection, where openset samples are semantically similar to the closed-set classes. Our work extends beyond near-OOD detection, which not only detects a near-OOD sample but also identifies its closest parent in the hierarchy of the closed-set classes.

**Hierarchical classification.** Based on how the work leverages the hierarchical structure of data, it can be divided into two types: (i) modify network architecture, and (ii) Constraint on the loss function. First line of work (Cerri et al., 2014; Chen et al., 2020a; Peng et al., 2018; Du et al., 2020; Zhao et al., 2021; Chen et al., 2019) modifies the network architecture to adapt the hierarchical structure of data. Among the state-of-the-art, Zhao et al. (Zhao et al., 2021) uses higher-order attention to discover relations in the form of a graph of embeddings. Similarly, Du et al. (Du et al., 2020) adopts progressive training and jigsaw puzzles to focus on both fine-grained and coarse-grained features required to identify hierarchical features. Graph-Neural-Network is also leveraged to construct a graph between class and attributes (Chen et al., 2019). Another line of work (Giunchiglia & Lukasiewicz, 2020; Deng et al., 2014; Chen et al., 2022; Chang et al., 2021) imposes constraints on the loss function to ensure hierarchical relations are being followed. The goal of the work in this setting is to improve the performance of hierarchical classes seen in the training. ProTeCt (Wu et al., 2024) proposes a tree-cut loss function for taxonomic openset detection. This work deals with identifying samples at different granularity in the hierarchy, not the samples from the open-world. Li et al. leverages the hierarchical relationship to perform hierarchical classification at the pixel level. Different from ours, this work does not consider the open-world setting. To handle fine-grained openset detection, our method ensures hierarchical constraints while performing openset detection in the hierarchy.

**Fine-grained openset detection.** Fine-grained openset detection is relatively under-explored in the literature. There are two ways of dealing the problem: (i) Local Detector and (ii) Global Detector. For the former, a local openset detector can be applied at each non-leaf class of the closed-set hierarchy. For each local openset detector, (Lee et al., 2018) quantifies the confidence score based on KL divergence between the model and uniform probability distribution and (Wang et al., 2022) quantifies uncertainty based on fuzzy logic. The global detector trains a single classifier to identify the sample between closed-set and openset classes. Several strategies are proposed to leverage samples from closed-set classes as openset ones (Lee et al., 2018). (Ruiz & Serrat, 2022) trains prototypes of closed and open set classes using a cosine loss. E-HND (Pyakurel & Yu, 2024) leverages an evidential loss to further separate closed and open-set classes by allocating fine-grained evidences. In this work, we address the fundamental limitations of both local and global detectors as outlined in the introduction by learning novel state-based node representations from a class hierarchy to advance fine-grained openset detection. Another line of work, Generalized Category Discovery (GCD) (Rastegar et al., 2023) targets a related problem similar to FGOD of assigning openset samples to a specific category. FGOD leverages an existing hierarchical structure and de-

fines openset class for every non-leaf class in the hierarchy so that openset test samples can be assigned to those openset classes. GCD, on the other hand, does not leverage existing hierarchy but learns to categorize the classes into a hierarchical structure. When a hierarchy is not present for the dataset, GCD approaches can be applied to obtain the hierarchy before the FGOD methods are applied. In that sense, GCD and HND are complementary to each other.

## 3. Methodology

We first formally define the problem of fine-grained openset detection (FGOD) with a hierarchy of closed-set classes. We present a standard process that builds a local detector on each node and performs openset detection along with the hierarchy. We provide novel theoretical insight on why it leads to a sub-optimal training process even considering all the ancestor nodes on the path leading to the ground-truth leaf node. Such theoretical insight inspires the design of the state-based node representations from a hierarchy that guarantees an optimal FGOD training process.

### 3.1. Problem Formulation

Given a training dataset: $D_{\text{train}} = \{x_n, y_n\}_{n=1}^N$, a data sample $x_n$ belongs to one of the $K$ classes as $y_n \in \mathcal{Y} = \{1, 2, ...K\}$. For the test dataset $D_{\text{test}} = \{x_{n'}, y_{n'}\}_{n'=1}^{N'}$, the test label $y_{n'}$ may or may not belong to train label set $\mathcal{Y}$. In general openset detection, the goal is to identify whether a test sample is from the closed-set of classes and if so, assigns it into the correct class. It can be formally defined as $f : x_{n'} \to \mathcal{Y} \cup \{K + 1\}$, where label $K + 1$ refers to the openset class.

Openset detection becomes more challenging when the the semantic similarity between openset samples and closed-set classes increases. By leveraging the hierarchy obtained from the classes associated with labels $\mathcal{Y}$, we formulate the problem of Fine-Grained Openset Detection (FGOD) with a hierarchy. For openset test samples, the goal of FGOD is to identify the node at which the sample becomes openset. To formally define the problem, we first introduce a few notations. For a node $y$ in a hierarchy $\mathcal{H}$, we refer to its parents, children, descendants, ancestors, and siblings as $\mathcal{P}(y), \mathcal{C}(y), \mathcal{D}(y), \mathcal{A}(y)$ and $\mathcal{S}_b(y)$. The hierarchy contains leaf nodes $\mathcal{L}(\mathcal{H})$ and non-leaf nodes $\mathcal{N}_l(\mathcal{H})$. Next, we assume that each non-leaf node $y$ has an extra child $\mathcal{O}(y)$, denoting the openset child, along with its closed-set children $\mathcal{C}(y)$. The set of all the openset nodes from the hierarchy is $\mathcal{O}(\mathcal{H})$. Formally, the goal of FGOD can be defined as $h : x_{n'} \to \mathcal{Y} \cup \mathcal{O}(\mathcal{H})$.

### 3.2. Why is a Local Detector Sub-optimal for FGOD with a Hierarchy?

By leveraging the hierarchy of closed-set classes, a local openset detector can be built on each non-leaf node. Each

detector performs classification of child nodes while quantifying an openset score. If the score is higher than a predetermined threshold, openset is detected. Otherwise, the sample is classified as one of the child nodes. The process starts from the root node and stops if openset is detected or reaches the leaf node of the hierarchy. For model training, we define known $D_{lc}^k$ and openset $D_{lc}^o$ data samples for each classifier $lc$. If a training sample belongs to descendant classes of a non-leaf node $lc$, it is assigned to $D_{lc}^k$, otherwise to $D_{lc}^o$. Using $D_{lc}^k$, a loss function (eg, cross-entropy) can be defined to guide the model to correctly classify its child nodes. Some regularization term is usually included to ensure a high openset score for samples from $D_{lc}^o$. Additional details and an algorithm summarizing the detection process is presented in the Appendix.

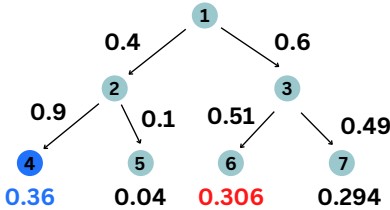

*Figure 3.* Local detector augmented with path probability

While using a sequence of local detectors offers an intuitive way to tackle FGOD, the final prediction depends on multiple classifiers. Each classifier makes a decision based on the local view only. If a single classifier makes a mistake, no matter how well other classifiers perform, the final prediction will be wrong. Thus, it fails to take advantage of fine-grained views from the hierarchy. As shown in Figure 3 for a test sample with ground-truth node ④. First, node 3 is predicted, instead of node ②, and then node 6 is predicted as the final prediction. However, it fails to examine that nodes ⑥ and ⑦ have a close to uniform distribution, while node 4 dominates the probability over node ⑤. If we could leverage fine-grained probabilities from all the descendants, this issue could be mitigated. An intuitive solution is to augment the local detector method by considering all the ancestor nodes of the ground-truth leaf node and maximize the the probability of the entire path. Specifically, for each leaf node in the hierarchy, we define a path from the root node to the leaf node. For each path, we calculate path probability by multiplying the local probabilities (conditioned on the parent node) of all the nodes that lie in the path.

$$\forall y \in \mathcal{L}(\mathcal{H}), \quad \texttt{ph}(y) = \prod_{a \in \mathcal{A}(y)} p(a|\mathcal{P}(a)) \qquad (1)$$

For each classifier, the local probability is obtained by the softmax operation on the logits of its child nodes. Therefore, the path probabilities are based on logits of only those nodes, whose parent classifiers are in the path. For example, for the training sample from node ④ in Figure 3, only logits

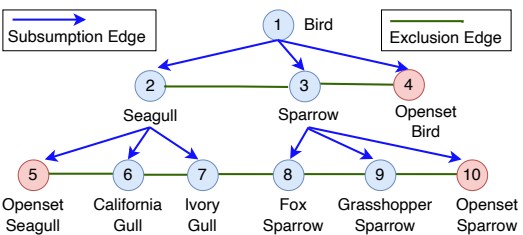

*Figure 4.* An example of open-set augmented hierarchy: Blue edge represents subsumption; Green edge represents exclusion.

| $v_1$ | $v_2$ | $v_3$ | $v_4$ | $v_5$ | $v_6$ | $v_7$ | $v_8$ | $v_9$ | $v_{10}$ | |
|---|---|---|---|---|---|---|---|---|---|---|
| 0 | 0 | 0 | 0 | 0 | 0 | 0 | 0 | 0 | 0 | $s_0$ |
| 1 | 0 | 0 | 1 | 0 | 0 | 0 | 0 | 0 | 0 | $s_4$ |
| 1 | 1 | 0 | 0 | 1 | 0 | 0 | 0 | 0 | 0 | $s_5$ |
| 1 | 1 | 0 | 0 | 0 | 1 | 0 | 0 | 0 | 0 | $s_6$ |
| 1 | 1 | 0 | 0 | 0 | 0 | 1 | 0 | 0 | 0 | $s_7$ |
| 1 | 0 | 1 | 0 | 0 | 0 | 0 | 1 | 0 | 0 | $s_8$ |
| 1 | 0 | 1 | 0 | 0 | 0 | 0 | 0 | 1 | 0 | $s_9$ |
| 1 | 0 | 1 | 0 | 0 | 0 | 0 | 0 | 0 | 1 | $s_{10}$ |

*Figure 5.* An example of global states

from nodes ②, ③, ④, and ⑤ are used in path probability calculation. Logits from the rest of the nodes are not used, making the optimization from path probability regularization sub-optimal, as shown in the following theorem.

**Theorem 3.1.** *(**Sub-optimality of FGOD with path probability**) Given a hierarchy $\mathcal{H}$ and a training sample $(x, y)$. $\mathcal{A}(y)$ is the set of nodes belonging to the path from root to leaf node $y$. Then, the cross-entropy loss function based on path probability does not guarantee a decrease in logits of node $y' \notin \mathcal{C}(a), \forall a \in \mathcal{A}(y)$.*

*Proof.* (Proof Sketch) We provide the detailed proof in Appendix C. First, we define path probability for ground truth label $y$ as a function of logits of the nodes in hierarchy, then evaluate the loss based on the probability of path $\text{ph}(y)$. Next, we calculate the gradient of the loss function for ground truth logit $g_y = \frac{d\mathcal{L}^{\text{ph}}}{dh(y|x)}$, and non-ground truth logits $g_{y'} = \frac{d\mathcal{L}^{\text{ph}}}{dh(y'|x)}$. We observe that gradient $g_y$ is dependent on logits of children of its parent only (local optimization). Similarly, for non-ground truth label $y' \notin \mathcal{C}(a), \forall a \in \mathcal{A}(y)$, the value of gradient $g_{y'} = 0$. There is no update for those non-ground truth logits. Therefore, there is no guarantee of a decrease in logits of node $y' \notin \mathcal{C}(a), \forall a \in \mathcal{A}(y)$. □

### 3.3. Learning State-Based Node Representations for Effective FGOD

To solve the problem of sub-optimality of FGOD using local detectors, it is essential to leverage all important hierarchical dependencies among the nodes so that the probability

optimization no longer considers local view only, and guarantees an update of all the parameters in the right direction. We denote $\mathcal{H}'$ as an augmented hierarchy $\mathcal{H}$ with openset nodes $\mathcal{O}(\mathcal{H})$ (see Figure 4 for an example). The leaf and non-leaf nodes of $\mathcal{H}'$ becomes $\mathcal{L}(\mathcal{H}') = \mathcal{L}(\mathcal{H}) \cup \mathcal{O}(\mathcal{H})$ and $\mathcal{N}_l(\mathcal{H}') = \mathcal{N}_l(\mathcal{H})$.

The hierarchy $\mathcal{H}'$ can be represented as a graph $G = (V, E_s, E_e)$ consisting of nodes $V = \{v_1, v_2, ... v_M\}$, where $M = |\mathcal{L}(\mathcal{H}') + \mathcal{N}_l(\mathcal{H}')|$. The relationship between two nodes $v_i$ and $v_j$ can be defined as an edge $(v_i, v_j)$, where each node is assigned the value of 1 or 0 to denote the node is either active or inactive in the edge. There are two types of edges in the graph: `subsumption` edges $E_s \subset V \times V$ and `exclusion` edges $E_e \subset V \times V$. A subsumption edge $(v_i, v_j) \in E_s$ is a directed edge that refers to node $v_i$ being a parent of $v_j$. Similarly, an exclusion edge $(v_i, v_j) \in E_e$ is an undirected edge that refers to nodes $v_i$ and $v_j$ being mutually exclusion to each other. The graphical nodes for the hierarchy of Bird along with subsumption and exclusion edges are illustrated in Figure 4. First, we define the legal assignment of the values for each edge, referred to as *local legal states*. Next, we define the legal assignment of the values to all the nodes, referred to as *global legal states*. The legal assignment for the $E_s$ is $(1, 1), (1, 0), (0, 0)$, meaning when the child node is active, the parent node has to be active. Similarly, the legal assignment for the $E_e$ is $(0, 1), (0, 0), (1, 0)$, meaning both of the nodes can not be active. The values except the legal assignments are illegal. With the definition of local legal states, we define a global legal state below.

**Definition 3.2** (Global Legal State). Given a graph $G = (V, E_s, E_e)$, a global legal state $s \in \{0, 1\}^M$ assigns values to all the nodes $V$ such that the following three conditions are satisfied: $C_1$: $\forall (v_i, v_j) \in E_s, (v_i, v_j) \neq (0, 1)$, $C_2$: $\forall (v_i, v_j) \in E_e, (v_i, v_j) \neq (1, 1)$, and $C_3$: $(\exists v \in \mathcal{L}(\mathcal{H}'), v = 1) \vee (\forall v, v = 0)$.

**Remark.** For each state, $C_3$ ensures that one of the leaf nodes is active or none of the nodes is active. Since we include openset nodes $\mathcal{O}(\mathcal{H})$ as leaf nodes of $\mathcal{H}'$, without the condition $(\exists v \in \mathcal{L}(\mathcal{H}')$, it will result in invalid global states. We also need to allow a state where the openset sample lies outside the entire hierarchy, which is given by $\forall v, v = 0$. All the legal global states can be represented in the form of a matrix, defined as global state matrix $\mathcal{S}$. The resulting dimension of $\mathcal{S}$ is $(|\mathcal{L}(\mathcal{H}')| + 1) \times M$. An example of the legal global states is presented in Figure 5, where nodes and states follow Figure 4. It is important to note that each state (except for $s_0$) corresponds to a leaf node in open-set augmented hierarchy $\mathcal{H}'$, so performing classification on the states is equivalent to conducting the multi-class classification as performed in the global detectors. More importantly, each state captures the important fine-grained dependencies among the ancestor nodes that

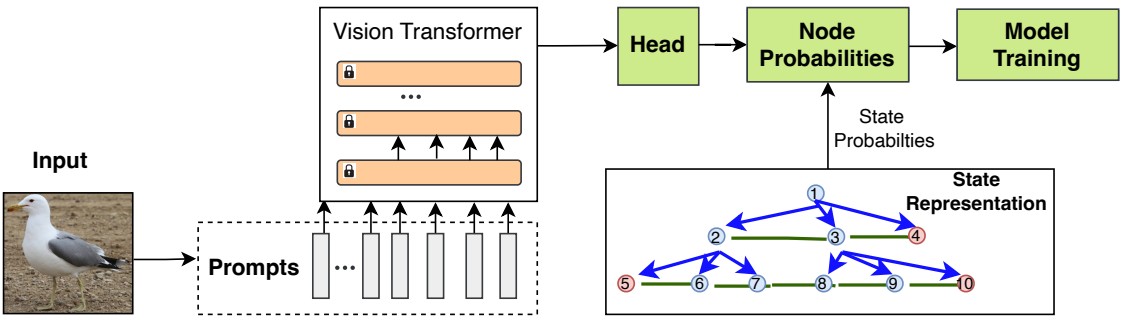

*Figure 6.* Conceptual diagram of model training State-FGOD

lead to the corresponding leaf node. If we look at state $s_6$, it corresponds to assigning a sample to California Gull, where nodes $v_1$, $v_2$, and $v_6$ are active and all the other nodes in the hierarchy are inactive. Those active nodes are ancestors of California Gull. Further, inactive nodes are siblings of either California Gull or its ancestors. Similar interpretations can be made about all the other states, where the index of the state corresponds to the node index from Figure 4 and the red color signifies openset cases.

**Evaluating the state and node probabilities.** Consider a network $h$ parameterized with $\theta$. We can obtain the logit from $h$ for all the nodes $V \in G$ represented as $h(v_1|x;\theta), ..., h(v_m|x;\theta), ...h(v_M|x;\theta)$. First, we calculate an un-normalized probability for each of the state $s$ by multiplying the exponent of logit from all the nodes that are active in state $s$:

$$\tilde{p}(s|x) = \prod_{m=1}^{M} e^{h(v_m|x;\theta)1_{(v_m=1|s)}} \qquad (2)$$

where $1_{(v_m=1|s)}$ is an indicator function that takes value of 1 when $v_m = 1$. The un-normalized state probabilities $\tilde{\mathbf{p}}$ is of dimension $|\mathcal{L}(\mathcal{H}')| + 1$, which can be normalized

$$Z = \sum_{s\in\mathcal{S}} \tilde{p}(s|x), \quad p(s|x) = \frac{\tilde{p}(s|x)}{Z} \qquad (3)$$

We can further obtain node probabilities $p(v_m|x)$ by summing all the state probabilities where the node $v_m$ is active:

$$p(v_m|x) = \sum_{s\in S} p(s|x)1_{(v_m=1|s)} \qquad (4)$$

**Proposition 3.3.** *For any node $v_m$, the parent-child relation satisfies the probability constraint:*

$$p_{v_m|x} \geq \sum_{c\in\mathcal{C}(v_m)} p(c|x),$$

$$p_{v_m|x} = \sum_{c\in\mathcal{C}(v_m)} p(c|x) + p(\mathcal{O}(v_m)|x) \qquad (5)$$

*With the complete knowledge of children of node $v_m$, the equality holds. Otherwise, the inequality indicates the presence of an openset node associated with $v_m$. In our formulation, we support the presence of an extra openset node.*

**Model training and inference.** We capture the hierarchical relationship between all the nodes through state representations. Next, we train a state-based detector that classifies all the leaf nodes in the hierarchy $\mathcal{H}'$. We remove one node $v_m$ at a time. This creates a dynamic hierarchy $\mathcal{H}' \setminus v_m$. When $v_m$ is removed, all the samples that belong to $v_m$ will be treated as openset samples with respect to $\mathcal{H}' \setminus v_m$. Since the openset samples can occur at different levels or positions within the class hierarchy, this process allows the model to learn to detect diverse types of openset samples, depending on their specific location within the hierarchy. It is designed to address the unique requirement for fine-grained open-set detection. For a training input $x$ with ground truth $y$, if $v_m = y$ or $v_m \in \mathcal{A}(y)$, then the ground truth label is changed to the openset class as $y = \mathcal{O}(\mathcal{P}(v_m))$. Otherwise, the ground truth label is not changed, and the sample belongs to closed-set classes. We train a single model parameterized by $\theta$. For each removed node $v_m$ and training sample $(x, y)$, the training loss maximizes ground truth node probability $p(y|x, \theta, \mathcal{H}' \setminus v_m)$. The loss function is:

$$\mathcal{L} = \mathop{\mathbb{E}}_{v_m \in V} \mathop{\mathbb{E}}_{(x,y)} [-\ln p(y|x, \theta, \mathcal{H}' \setminus v_m)] \qquad (6)$$

The conceptual diagram of the proposed method along with model training is illustrated in Figure 6. The inference process using the proposed State-FGOD is summarized in Algorithm 2 of the Appendix. Next, we show that minimization of the cross entropy loss function using state-based formulation leads to optimal model training in Theorem 3.4.

**Theorem 3.4.** *(**Optimal model training from state-based node representations**) Given a hierarchy $\mathcal{H}'$ and a training sample $(x, y)$, the ground truth logits are $h(a|x;\theta), \forall a \in \mathcal{A}(y)$ and the non-ground truth logits are $h(a'|x;\theta), \forall a' \notin$*

$\mathcal{A}(y)$. *Then, the cross-entropy loss function based on state probability guarantees the increase in value of ground truth logits and decrease in the value of non-ground truth logits.*

*Proof.* (Proof Sketch) First, we define cross-entropy loss $\mathcal{L}$ based on node probability $p(y|x)$. Next, we calculate the gradient of the loss function for the ground truth logit $g_y = \frac{d\mathcal{L}}{dh(y|x)}$, and non-ground truth logits $g_{y'} = \frac{d\mathcal{L}}{dh(y'|x)}$. We observe that gradient $g_y$ is dependent on logits of all nodes (global view). When logit of non-ground truth increases, the gradient magnitude increases, updating the model parameters such that it increases the logit of ground truth node. Similarly, for non-ground truth label $y'$, the gradient $g_{y'}$ updates the model parameters such that it decreases the value of the non-ground truth logit $h(y'|x)$. Therefore, this guarantees a decrease in logits of non-ground truth node $y'$. The detailed proof is in Appendix C. $\square$

Along with the proofs, we have further illustrated the concept of gradient analysis of Theorems 3.1 and 3.4 using the representative example from Figure 4 in Appendix C.3. As pointed out in the introduction, building a global detector by performing multi-class classification over all the leaf nodes in $\mathcal{H}'$ may lead to a high hierarchical detection error. The following Corollary provides the root cause for that, which explains why a global detector is sub-optimal.

**Corollary 3.5** (Sub-optimality of global detectors). *Given a hierarchy $\mathcal{H}'$ and a training sample $(x, y)$, the ground truth labels are $a \in \mathcal{A}(y)$ and the non-ground truth labels are $a' \notin \mathcal{A}(y)$. The cross-entropy loss based on probability from only leaf labels, $y$ and $y' \notin \mathcal{A}(y) \wedge y' \in \mathcal{L}(\mathcal{H}')$ does not increase the value of non-leaf ground truth logits.*

**Remarks.** Existing global detector methods do not fully leverage fine-grained hierarchical dependencies to train the model, and hence non-leaf logits are not optimized for fine-grained openset detection. As a result, these methods may assign the openset sample to the incorrect parent node, leading to lower hierarchical precision as shown in Figure 2.

## 4. Experiments

In this section, we first present the datasets and baselines used in the experiments. Next, we discuss the comparison settings and evaluation metrics. We then show the FGOD results and ablation studies to justify the proposed method.

**Datasets and implementation details** We use the following real-world hierarchical datasets for the experiments. (i) Tiny Imagenet (Le & Yang, 2015): a subset of ImageNet, (ii) CUB-200-2011 (Welinder et al., 2010): images of fine-grained species of bird (iii) Animals With Attributes 2 (AWA2): images of animals (Lampert et al., 2014). Details for each dataset are presented in Appendix D. For both the compared algorithms and the proposed method, we experiment with ViT-B/16 using prompt-based parameter-efficient

fine-tuning. Additionally, we conduct an ablation study to evaluate different backbone architectures and also compare different fine-tuning methods.

**Baselines.** There are two categories of methods that can be adapted for FGOD: (i) Local detector and (ii) Global detector methods. For local detector, the following baselines can be leveraged: top-down (TD), maximum softmax probability (MSP) and hierarchical classification with extra Node (HC-EN). For global detector, the following baselines are compared: Dual Accuracy Reward Trade-off Search (DARTS) (Deng et al., 2012), Relabel (Lee et al., 2018), Leave-One-Out (LOO) (Lee et al., 2018), Evidential learning (Sensoy et al., 2018; Pyakurel & Yu, 2024), TD+LOO (Lee et al., 2018) and E-HND (Pyakurel & Yu, 2024). The details are presented in Appendix.

### 4.1. Comparison Settings and Evaluation Metrics
We compare the performance of the proposed method with the baselines using top-1 accuracy. For an openset sample $\{x_{n'}, y_{n'}\}$ with prediction $\hat{y}_{n'}$, it is correct if $\hat{y}_{n'} = \mathcal{O}(\mathcal{P}(y_{n'}))$. The number of correct predictions out of closed-set samples is called Closed-set Accuracy (CA), and openset samples are Openset Accuracy (OA). For global detector methods, we can adjust the bias to the logits of the leaf and non-leaf nodes to obtain different CA and OA. We present the comparison of global detector methods with AUC and OA at $50\%$ CA as $OA@50$.

### 4.2. Fine-Grained Openset Detection Results
We present the openset accuracy and AUC results in Table 1. The comparison reveals the superior performance of our method. We observe that the global detector-based methods, in general, have better performance than local detector-based methods. In Tiny Imagenet, the performance gap is more prominent as the dataset has 12 levels in the hierarchy, and the error accumulation from the detectors becomes more severe. Among local detector methods, the elimination of setting thresholds on openset score improves the performance, as given by HC-EN baseline for the Tiny Imagenet dataset. For other datasets, local detector methods based on different openset score quantification methods, MSP and KL divergence score in TD method, also show competitive performance. This suggests that the performance of the local detector-based method is impacted by the number of hierarchy levels, quality of openset scores, and thresholds on the openset scores. In general, due to the decision of local classifiers, the local detector methods achieve lower FGOD performance in comparison to global detector methods. For global detector methods, the proposed State-FGOD consistently outperforms other methods on all the datasets (improvement in AUC of CUB:$+5.45$, Tiny Imagenet:$+2.45$ and AWA2:$+2.82$ w.r.t. the best-performing baseline). This is further demonstrated by AUC curve Figure 7. Among the baselines, LOO, and E-HND methods demonstrate competi-

*Table 1.* Comparison with Local and Global Detectors

| Method | CUB | | Tiny Imagenet | | AWA2 | |
|---|---|---|---|---|---|---|
| | **CA/OA** | **HMean/AUC** | **CA/OA** | **HMean/AUC** | **CA/OA** | **HMean/AUC** |
| **(a) Local Detector Methods** | | | | | | |
| TD | 72.47/18.75 | 29.79/− | 50.43/9.20 | 15.56/− | 80.42/33.83 | 47.62/− |
| MSP | 72.62/24.23 | 36.33/− | 62.68/7.52 | 13.42/− | 89.12/30.28 | 45.19/− |
| HC-EN | 78.26/8.30 | 15.01/− | 48.83/11.56 | 18.69/− | 89.77/26.08 | 40.41/− |
| **(b) Global Detector Methods** | | | | | | |
| DARTS | 50.00/46.74 | 48.32/41.82 | 50.00/17.15 | 25.54/12.75 | 50.00/38.29 | 43.36/34.29 |
| Relabel | 50.00/50.95 | 50.47/41.35 | 50.00/22.02 | 30.58/17.58 | 50.00/30.50 | 39.52/32.68 |
| Evidential | 50.00/43.34 | 46.43/33.41 | 50.00/19.35 | 27.90/14.53 | 50.00/46.95 | 48.42/36.93 |
| LOO | 50.00/48.34 | 49.15/38.45 | 50.00/21.53 | 30.10/17.26 | 50.00/48.83 | 49.40/45.31 |
| TD+LOO | 50.00/44.17 | 46.90/32.77 | 50.00/19.37 | 27.92/14.87 | 50.00/35.26 | 41.35/28.40 |
| E-HND | 50.00/54.43 | 52.12/45.59 | 50.00/21.45 | 30.02/17.31 | 50.00/47.05 | 48.47/41.69 |
| **State-FGOD** | **50.00/64.47** | **56.31/51.04** | **50.00/24.71** | **33.07/ 20.03** | **50.00/ 53.32** | **51.60 /48.13** |

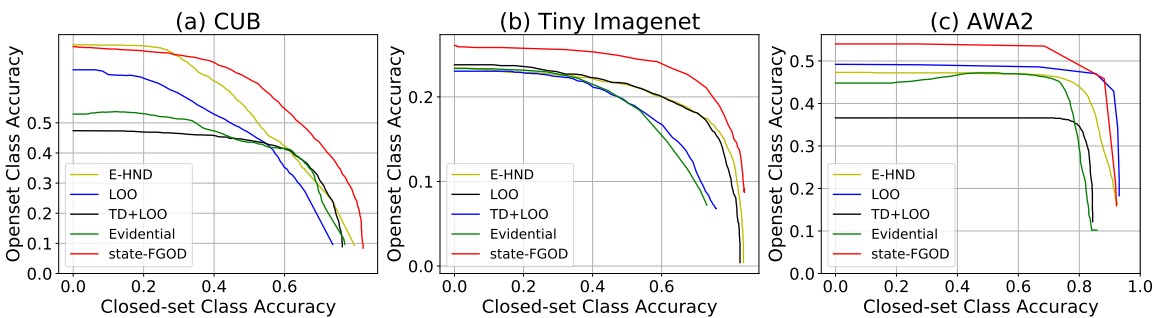

*Figure 7.* Closed-set accuracy vs Openset accuracy curves

*Table 2.* Comparison using hierarchical precision and recall

| Method | Hierarchical Precision | | Hierarchical Recall | | HarmonicMean | |
|---|---|---|---|---|---|---|
| | **Closed-set** | **Openset** | **Closed-set** | **Openset** | **Closed-set** | **Openset** |
| TD | 96.88 | 90.36 | 84.69 | 87.05 | 90.37 | 88.67 |
| LOO | 95.96 | 83.86 | 87.22 | 92.31 | 91.38 | 87.88 |
| TD+LOO | 94.67 | 84.75 | 85.03 | 86.43 | 89.59 | 85.58 |
| E-HND | 97.23 | 85.83 | 86.36 | 92.07 | 91.47 | 88.84 |
| State-FGOD | **98.06** | **90.68** | **87.40** | **92.81** | **92.42** | **91.73** |

tive performance.

**Hierarchical performance measure.** We conduct this study to understand the performance of our method w.r.t hierarchical performance measure. We use the CUB dataset with hierarchical metrics: hierarchical precision, hierarchical recall, and harmonic mean. Apart from leaf-level ground truth, hierarchical metrics also measure whether the ancestors of the ground truth leaf node are being correctly predicted. `Hierarchical Precision (HP)` measures the number of nodes correctly predicted by the model out of all the nodes predicted in the hierarchy. Similarly, `Hierarchical Recall (HR)` measures the number of nodes correctly predicted out of all the ground truth nodes

from the hierarchy. For a sample $(x, y)$, if the prediction is $\hat{y}$, the hierarchical metrics are defined as:

$$\texttt{HP} = \frac{|\texttt{Common Ancestors}(y, \hat{y})|}{|\texttt{Ancestors}(\hat{y})|}, \quad (7)$$

$$\texttt{HR} = \frac{|\texttt{Common Ancestors}(y, \hat{y})|}{|\texttt{Ancestors}(y)|} \quad (8)$$

The comparison table using hierarchical metrics is presented in Table 2. Along with our method, we compare performance of TD method (a representative method from local detectors) and several representative methods from global detectors. As can be seen, local detector has poor openset recall. The error accumulation of local detector affects the

Table 4. Impact of state-based representation

| Method | SP? | OA@50 | AUC |
|--------|-----|-------|-----|
| DARTS | × | 46.74 | 41.82 |
| | ✓ | $57.02_{(+10.28)}$ | $46.71_{(+4.89)}$ |
| Relabel | × | 50.95 | 41.35 |
| | ✓ | $59.80_{(+8.85)}$ | $45.71_{(+4.36)}$ |
| LOO | × | 48.34 | 38.45 |
| | ✓ | $55.31_{(+6.97)}$ | $46.23_{(+7.78)}$ |

Table 6. Impact of path probability

| | PP? | TD | MSP | HC-EN |
|---|-----|----|----|-------|
| CA | × | 72.47 | 72.62 | 78.26 |
| | ✓ | $72.63_{(+0.16)}$ | $72.17_{(-0.45)}$ | $77.58_{(-0.68)}$ |
| OA | × | 18.75 | 24.23 | 8.30 |
| | ✓ | $21.16_{(+2.41)}$ | $28.27_{(+4.04)}$ | $10.05_{(+1.75)}$ |
| HMean | × | 29.79 | 36.33 | 15.01 |
| | ✓ | $32.77_{(+2.98)}$ | $40.62_{(+4.29)}$ | $17.79_{(+2.78)}$ |

recall performance, as the ground truth node lies in leaf nodes of hierarchy. On the other hand, global detector doesn't fully utilize the fine-grained relationships in the hierarchy resulting in poor openset precision. The proposed State-FGOD is able to avoid error accumulation while effectively capturing important dependencies among different nodes in hierarchy, which achieves a good balance on both hierarchical metrics for closed-set and openset classes.

**Top-2 Accuracy.** Table 3 compares representative global detectors using top-2 accuracy on the CUB dataset. The performance improvement shows the effectiveness of the proposed method. By leveraging the relationships among all the nodes, it effectively improves the performance of FGOD.

Table 3. Top-2 Accuracies

| Method | OA@50 | AUC |
|--------|-------|-----|
| LOO | 70.08 | 62.49 |
| TD+LOO | 57.94 | 48.39 |
| EHND | 84.58 | 74.08 |
| State-FGOD | **87.24** | **76.72** |

### 4.3. Ablation Studies

**Impact of state-representation in global detectors.** We study how the state formulation impacts the global detector methods. We apply state formulation on representative methods: DARTS, Relabel, and LOO on CUB dataset and present our results in Table 4. Using state formulation in global detector methods results in significant performance improvement. This study further highlights the importance of the utilization of hierarchy relationships between nodes.

**Impact of backbone.** We study the impact of different backbones to the proposed method. We experiment with CUB dataset and architectures: ResNet101, ResNet152 (He et al., 2016), and ViT-B/16. For ResNet-based methods, we use a linear layer on top of Resnet features for fine-tuning. For ViT, we leverage popular parameter-efficient fine-tuning methods: prompt (Jia et al., 2022), bias (Cai et al., 2020), and adapter (Rebuffi et al., 2017). The results are presented in Table 5. Among different settings, ViT-B/16 fine-tuned with prompt, leads to superior performance. Prompt fine-tuning allows training learnable parameters in the input

Table 5. Impact of backbone

| Backbone | OA@50 | AUC |
|----------|-------|-----|
| ResNet101 | 45.63 | 34.39 |
| ResNet152 | 44.34 | 33.73 |
| ViT-B/16 + Adapter | 59.58 | 47.12 |
| ViT-B/16 + Bias | 58.45 | 47.72 |
| ViT-B/16 + Prompt | **64.47** | **51.04** |

space, which helps learn hierarchical relationships between classes in the input space.

**Impact of component:** The conceptual diagram of the proposed method is presented in Figure 6. To better learn the hierarchical relationships, we have two components in the proposed method: (i) Prompt Tuning (ii) State representation. We study the impact of each component in CUB dataset. We present the results in Table 7. As we remove each component, we see a significant drop in the performance, justifying the importance of the components.

Table 7. Impact of component

| Method | OA@50 | AUC |
|--------|-------|-----|
| w/o Prompt | 55.31 | 46.23 |
| w/o State | 48.34 | 38.45 |
| State-FGOD | **64.47** | **51.04** |

**Impact of path probability.** In our methodology, we discuss how local detector methods are not able to leverage the fine-grained probability information to make decisions leading to error accumulation. We use path probability to identify the path in a hierarchy instead of all the classifier's predictions. For each local detector method, we compare the performance with and without path probability for CUB dataset in Table 6. With the use of path probability, we see improvement of both closed-set and openset performance in most of the methods. This study shows the importance of utilization of fine-grained probabilities for FGOD.

## 5. Conclusion

Openset detection poses a significant challenge when the samples are semantically similar to the closed-set classes. In this work, we leverage the hierarchical structure of the closed-set classes to detect the openset samples in a fine-grained manner. We formulate legal states to fully capture the hierarchical relationship between all the nodes in the hierarchy that allows us to optimize the performance of recognizing openset samples at different levels in the hierarchy. Further, we perform theoretical analysis to show the fundamental advantage of the proposed method. Our approach exhibits superior performance, as validated through extensive experimentation on various real-world hierarchical datasets.

## Impact Statement

In this work, we leverage the hierarchical relationships to perform fine-grained openset detection. Such hierarchical relations are useful by providing more information about the openset data samples that leads to an informed decision. The proposed approach can be potentially applied in many fields like wildlife monitoring, studying the nature of malware attacks, and identifying the parent class of traffic signs for self-driving cars.

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

# Learning State-Based Node Representations from a Class Hierarchy for Fine-Grained Open-Set Detection

## Supplementary Material

**Organization.**   In section A, we present the summary of all the notations used in the paper. In section B, we discuss the details of local and global detectors. In section C, we provide the proofs and illustrations of theorems presented in the main paper. In section D, we discuss details of experiments and provide additional results. In section E, we discuss potential societal impacts of the work, along with the limitation and future works. In section F, we provide the source code.

## A. Summary of Notations

| Notation | Description |
|:---:|:---:|
| $\mathcal{H}$ | A training hierarchy of closed-set classes |
| $D_{\text{train}}$ | Training Dataset |
| $N$ | Total number of data samples in training |
| $\mathcal{Y}$ | Set of all leaf labels of training dataset |
| $\mathcal{L}(\mathcal{H})$ | Set of all leaf nodes; is a function of hierarchy |
| $\mathcal{N}_l(\mathcal{H})$ | Set of all non-leaf nodes; is a function of hierarchy |
| $\mathcal{P}(y)$ | Parents of node $y$ |
| $\mathcal{C}(y)$ | Children of node $y$ |
| $\mathcal{D}(y)$ | Descendants of node $y$ |
| $\mathcal{A}(y)$ | Ancestors of node $y$ |
| $\mathcal{S}_b(y)$ | Siblings of node $y$ |
| $D_{\text{test}}$ | Test Dataset |
| $\mathcal{O}(y)$ | An openset class belonging to $y$ |
| $\mathcal{O}(\mathcal{H})$ | A set of openset classes of hierarchy $\mathcal{H}$ |
| $\mathcal{H}'$ | A resulting hierarchy of $\mathcal{H}'$ augmented with $\mathcal{O}(\mathcal{H})$ |
| $D_{lc}^k$ | Known dataset for classifier $lc$ |
| $D_{lc}^o$ | Openset dataset for classifier $lc$ |
| $\text{ph}(y)$ | Path probability of path from root to node $y$ |
| $g_y$ | Gradient of loss w.r.t ground truth logit |
| $g_{y'}$ | Gradient of loss w.r.t non-ground truth logit |
| $G$ | Graph representing hierarchy $\mathcal{H}'$ |
| $V$ | All the nodes of the graph $G$ |
| $E_s$ | A set of all the subsumption edges in the graph $G$ |
| $E_e$ | A set of all the exclusion edges in the graph $G$ |
| $s$ | A global state assigning binary values to all the nodes in $G$ |
| $\mathcal{S}$ | Global state matrix of $G$ |
| $M$ | Total number of nodes in $G$ |
| $h$ | FGOD network |
| $h(v_m|x;\theta)$ | Logit from FGOD network for node $v_m$ |
| $\tilde{p}(s|x)$ | Un-normalized probability for state $s$ |
| $p(s|x)$ | Normalized probability for state $s$ |
| $p(v_m|x)$ | Probability for node $v_m$ |
| $\theta_{\mathcal{H}'}$ | FGOD network parameters |
| $\theta_{\mathcal{H}'\setminus a}$ | FGOD network parameters when node $a$ is removed from $\mathcal{H}'$ |

## B. FGOD using Local and Global Detectors

In this section, we provide additional details on using local and global detectors for FGOD.

---

**Algorithm 1** FGOD with a hierarchy

---

   Require: A set of thresholds $\boldsymbol{\lambda}$ for all local detectors (one for each non-leaf node).
   Current local classifier, $lc \leftarrow$ root node
  **while** Stopping Condition $\neq 1$ **do**
     Probability distribution of $lc$: $\boldsymbol{p} \in \mathbb{R}^{|\mathcal{C}(lc)|}$
     Openset score: $o \in \mathbb{R}$
     Prediction from $\hat{y}$: $argmax(\boldsymbol{p})$
     **if** $o \leq \lambda_{lc}$ **then**
       $lc$: predicted child node, $\hat{y}$
       **if** $lc \in \mathcal{L}(\mathcal{H})$ **then**
         Final Prediction $\leftarrow lc$, Stopping Condition $\leftarrow 1$
       **end if**
     **else**
       Final Prediction $\leftarrow \mathcal{O}(lc)$, Stopping Condition $\leftarrow 1$
     **end if**
  **end while**

---

**Local detector based FGOD.** Algorithm 1 summarizes the open-set detection process using a local detector. Based on how openset is determined, existing general openset detection methods can be adapted for FGOD. Here, we consider some commonly used open-detection strategies. For example, the openset score can be quantified using the KL divergence between a uniform distribution and model predicted distribution over the child nodes (eg, the top-down method (Lee et al., 2018)), model predicted maximum softmax probability (MSP) (Vaze et al., 2021), free energy based evaluation (Liu et al., 2020), or evidential uncertainty (Sensoy et al., 2018). In what follows, we first use the top-down method as a concrete example to present a specific loss function in (9) and then generalize into a more abstract form in (10):

$$\mathcal{L}^{\mathcal{H}} = \mathop{\mathbb{E}}_{lc \in \mathcal{N}_l(\mathcal{H})} \Big\{ \mathop{\mathbb{E}}_{(x,y) \in D_{lc}^k} [-\ln p(y|x, lc; \theta)] + \mathop{\mathbb{E}}_{(x,y) \in D_{lc}^o} KL[U(.|lc)||p(.|x, lc; \theta)] \Big\} \tag{9}$$

$$= \mathop{\mathbb{E}}_{lc \in \mathcal{N}_l(\mathcal{H})} \{ \mathcal{L}_{lc}^1 + \mathcal{L}_{lc}^2 \} \tag{10}$$

where $\mathcal{L}_{lc}^1$ maximizes the probability of ground truth class for each classifier using known samples $D_{lc}^k$, and $\mathcal{L}_{lc}^2$ is a regularization term for the classifier to assign a high openset score to openset samples in $D_{lc}^o$.

We summarize the inference process of FGOD with a hierarchy in Algorithm 1.

In addition to using the KL divergence to specify the confidence, we may consider other options, including (i) Maximum Softmax Probability (MSP) (Vaze et al., 2021), evidential uncertainty (Sensoy et al., 2018), energy score (Liu et al., 2020), and Maximum Logit (Vaze et al., 2021) as openset score. For MSP baseline, the training is the same as the Top-down method, as in Eq. (9). For other baselines, we use the loss function according to evidential loss, where the first loss is to output high evidence for the correct class, and KL regularizer is used to train the novel samples to output uniform KL divergence. The evidential based loss function is:

$$\mathcal{L}^{\text{HC-evidential}} = \mathop{\mathbb{E}}_{lc \in \mathcal{N}_l(\mathcal{H})} \Big\{ \mathop{\mathbb{E}}_{(x,y) \in D_{lc}^k} \sum_{k=1}^{|\mathcal{C}(lc)|} y_{ik}[\ln(St_i^{lc}) - \ln(\alpha_{ik})]$$
$$+ \mathop{\mathbb{E}}_{(x,y) \in D_{lc}^o} KL[D(.|x, lc; \theta)||D(.| < 1, 1, ..., >, lc)] \tag{11}$$

For each method, if the following conditions hold, the sample is not detected as openset.

$$\text{Maximum Softmax Probability: } \max(Pr(.|x, lc; \theta)) \geq \lambda_{lc} \tag{12}$$

$$\text{Evidential Uncertainty: } \frac{|\mathcal{C}(lc)|}{St_i^{lc}} \leq \lambda_{lc} \tag{13}$$

$$\text{Energy Score: } -\ln \sum_{k=1}^{|\mathcal{C}(lc)|} \exp h_k(x; \theta) \leq \lambda_{lc} \tag{14}$$

$$\text{Maximum Logit Score: } \max(h_k(x; \theta)) \geq \lambda_{lc} \tag{15}$$

Instead of relying on the openset score, samples in $D_{lc}^o$ can be treated as an extra openset child node $\mathcal{O}(y)$ (Neal et al., 2018). In this way, if the classifier predicts the highest probability for $\mathcal{O}(y)$, it detects the sample as openset. Figure 8 provides a specific example on how to add an extra node to each non-leaf node in the hierarchy that transforms open-set detection in a hierarchy into a multi-class classification problem, where each leaf node correspond to a class. In this case, the openset score comparison no longer depends on a pre-defined threshold. If the highest probability of any child node is lower than that of the extra novel node, we can classify the sample as openset. A loss function can be defined accordingly:

$$\mathcal{L}^{\mathcal{H}} = \underset{lc \in \mathcal{N}_l(\mathcal{H})}{\mathbb{E}} \left\{ \underset{(x,y) \in D_{lc}^k}{\mathbb{E}} [-\ln p(y|x, lc; \theta)] + \underset{(x,y) \in D_{lc}^o}{\mathbb{E}} [-\ln p(\mathcal{O}(lc)|x, lc; \theta)] \right\} \tag{16}$$

Similarly, each detector leverages the openset score to find out if the sample is openset or not. For this method, if the following condition holds, the sample is not openset.

$$\max(\mathbf{p}(.|x, lc; \theta)) \geq p(\mathcal{O}(lc)|x, lc; \theta) \tag{17}$$

**Details of Global Detector Methods**

- Dual Accuracy Reward Trade-off Search (DARTS) (Deng et al., 2012) is adapted for FGOD by training a classifier using leaf nodes and obtaining the expected rewards for all the leaf nodes and openset non-leaf nodes.

- Relabel (Lee et al., 2018) trains a single classifier using leaf and openset non-leaf nodes by relabeling some of the leaf nodes as openset ancestor nodes.

- Leave-One-Out (LOO) (Lee et al., 2018) trains a classifier with leaf and openset non-leaf nodes. For openset nodes, each node from the hierarchy is removed one at a time to use it as an openset parent node.

- Top Down + Leave-One-Out (TD+LOO) (Lee et al., 2018) uses input from Top Down method to LOO training.

- Evidential learning (Sensoy et al., 2018; Pyakurel & Yu, 2024) uses the evidential loss to train a classifier following the LOO strategy. Evidential uncertainty is used for openset detection.

- Evidential-Hierarchical Novelty Detection (E-HND) (Pyakurel & Yu, 2024) trains the model to separate evidence margin of closed-set and openset classes.

**Open-set Detection using State-FGOD.** The inference process of State-FGOD is summarized in Algorithm 2.

## C. Proofs of Theorems

### C.1. Proof of Theorem 1

*Proof.* Let us assume the ground truth node as $y$. For simplicity purpose, we denote the logit output by the neural network by $o$. For example: we denote $h(y|x)$ by $o_y$. The path probability of the node $y$ is given by:

$$\text{ph}(y|x) = \prod_{a \in A(y)} p(a|P(a); x) \tag{18}$$

$$= \frac{e^{o_y}}{e^{o_y} + \sum_{sb \in \mathcal{S}_b(y)} e^{o_{sb}}} \times \frac{e^{o_{\mathcal{P}(y)}}}{e^{o_{\mathcal{P}(y)}} + \sum_{sb' \in \mathcal{S}_b(\mathcal{P}(y))} e^{o_{sb'}}} \times ... \times 1 \tag{19}$$

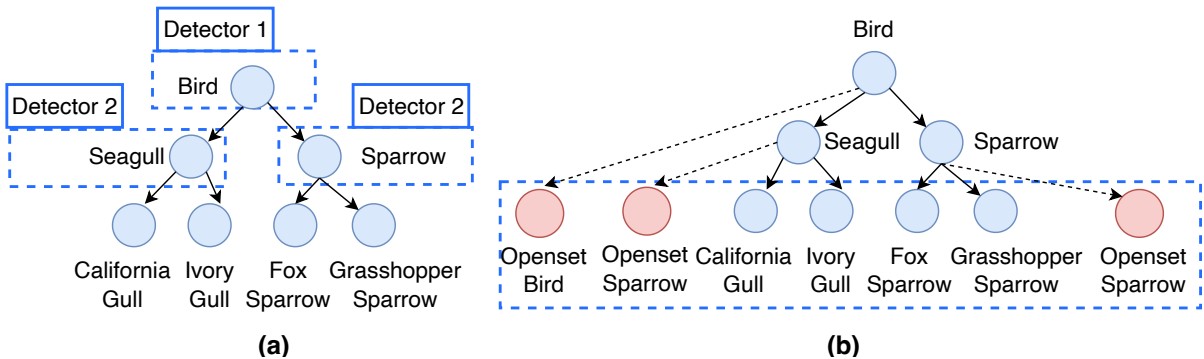

*Figure 8.* A diagrammatic representation of (a) Local detector method: There are three classifiers on Bird, Seagull and Sparrow, respectively. (b) Global Detector method: A single global classifier for all leaf nodes and openset non-leaf nodes.

---

**Algorithm 2** Inference using State-FGOD

---

`Require`: A set of biases $\mathbf{b}$.
`Require`: A trained State-FGOD model $h$ parameterized with $\theta$.
`Input`: A test data sample $\{x_{n'}, y_{n'}\}$.
Logits for all the $M$ nodes: $h(v_1|x_{n'}; \theta), h(v_2|x_{n'}; \theta), ..., h(v_m|x_{n'}; \theta), ...h(v_M|x_{n'}; \theta)$
Un-normalized probabilities for each state $\tilde{p}(s|x_{n'})$ using Eq. (2).
Un-normalized probabilities for each leaf node $v_m \in \mathcal{L}(\mathcal{H}')$
$\quad \tilde{p}(v_m|x_{n'}) = \sum_{s \in S} \tilde{p}(s|x_{n'})1_{(v_m=1|s)}.$
**for** $b$ in biases $\mathbf{b}$ **do**
$\quad$ Add bias to logits of openset nodes, $\forall v_m \in \mathcal{O}(\mathcal{H}), \quad \tilde{p}(v_m|x_{n'}) = \tilde{p}(v_m|x_{n'}) + b$
$\quad$ Obtain normalized probabilities for all leaf nodes of $\mathcal{H}'$ as $\boldsymbol{p}$
$\quad$ Select node with the highest probability as $\hat{y} = argmax(\boldsymbol{p})$
**end for**

---

The definition of cross-entropy loss for path probability becomes.

$$\mathcal{L}^{\text{ph}} = -\log\left[\frac{e^{o_y}}{e^{o_y} + \sum_{sb \in \mathcal{S}_b(y)} e^{o_{sb}}} \times \frac{e^{o_{\mathcal{P}(y)}}}{e^{o_{\mathcal{P}(y)}} + \sum_{sb' \in \mathcal{S}_b(\mathcal{P}(y))} e^{o_{sb'}}} \times ... \times 1\right] \quad (20)$$

First, we take the derivative of loss w.r.t ground truth logit $o_y$. All the terms except the $1^{st}$ one are not a function of $o_y$, so we treat them as a constant $con$. Similarly, we denote the normalization of the first fraction as $Z_y$.

$$\frac{d\mathcal{L}^{\text{ph}}}{do_y} = \frac{d}{do_y} - \log\left[\frac{e^{o_y}}{e^{o_y} + \sum_{sb \in \mathcal{S}_b(y)} e^{o_{sb}}} \times con\right] \quad (21)$$

Using chain and product rule of derivative

$$= -\frac{Z_y}{e^{o_y}} \frac{1}{con} \times \left[\frac{e^{o_y}}{Z_y} \times 0 + con \times \frac{Z_y e^{o_y} - e^{o_y} e^{o_y}}{Z_y^2}\right] \quad (22)$$

$$= -\frac{Z_y}{e^{o_y}} \times \frac{1}{con} \times \frac{con}{Z_y^2} \times \left[e^{o_y}(Z_y - e^{o_y})\right] \quad (23)$$

$$= -\left[1 - \frac{e^{o_y}}{e^{o_y} + \sum_{sb \in \mathcal{S}_b(y)} e^{o_{sb}}}\right] \quad (24)$$

The gradient is negative, meaning minimizing the loss function increases the ground truth logit $o_y$. If the value of $o_y$ is low, the gradient magnitude is high. Similarly, for a high value of non-ground truth sibling leaf logit, $o_{sb}, sb \in \mathcal{S}_b(y)$, the gradient magnitude is high. Except for the non-ground truth sibling leaf logits, the gradient magnitude is not dependent

on logits of other nodes, which implies a local optimization. Next, we take derivative w.r.t a non-ground truth leaf logit, $sb \in \mathcal{S}_b(y)$. The terms other than the $1^{st}$ are not the function of $sb$, so we treat them as a constant $con$.

$$\frac{d\mathcal{L}^{\text{ph}}}{do_{sb}} = \frac{d}{do_{sb}} - \log\left[\frac{e^{o_y}}{e^{o_y} + \sum_{sb\in\mathcal{S}_b(y)} e^{o_{sb}}} \times con\right] \tag{25}$$

Using chain and product rule of derivative

$$= -\frac{Z_y}{e^{o_y}}\frac{1}{con} \times \left[\frac{e^{o_y}}{Z_y} \times 0 + con \times \frac{Z_y \times 0 - e^{o_y}e^{o_{sb}}}{Z_y^2}\right] \tag{26}$$

$$= \frac{Z_y}{e^{o_y}} \times \frac{1}{con} \times \frac{con}{Z_y^2} \times \left[e^{o_y} \times e^{o_{sb}}\right] \tag{27}$$

$$= \frac{e^{o_{sb}}}{e^{o_y} + \sum_{sb\in\mathcal{S}_b(y)} e^{o_{sb}}} \tag{28}$$

The gradient is positive, meaning minimizing the loss function decreases the non-ground truth sibling leaf logit. If the value of $o_{sb}$ is high, the gradient magnitude is high. Similar gradient analysis can be obtained for all non-ground truth nodes, $y' \in \mathcal{C}(a), \forall a \in \mathcal{A}(y)$. Next, we take derivative w.r.t a non-ground truth logit of the node $y' \notin \mathcal{C}(a), \forall a \in \mathcal{A}(y)$, which are not child of the node in the path of the root node to the ground truth node.

$$\frac{d\mathcal{L}^{\text{ph}}}{do_{y'}} = 0 \tag{29}$$

The gradient is $0$ as the loss is not a function of the $y'$. So, minimizing the loss function does not update the logits of these nodes. Therefore, it does not guarantee the decrease of all the non-ground truth logits. $\qquad\square$

### C.2. Proof of Theorem 2

*Proof.* The state probability of the node $y$ is given by:

$$\tilde{p}(y|x) = \prod_{a\in A(y)} e^{o_a} = e^{\sum_{a\in A(y)} o_a} \tag{30}$$

$$p(y|x) = \frac{\tilde{p}(y|x)}{1 + \sum_{l\in\mathcal{L}(\mathcal{H}')} \tilde{p}(l|x)} \tag{31}$$

$$= \frac{e^{\sum_{a\in A(y)} o_a}}{1 + \sum_{l\in\mathcal{L}(\mathcal{H}')} e^{\sum_{a'\in A(l)} o_{a'}}} \tag{32}$$

The cross-entropy loss for state probability becomes

$$\mathcal{L} = -\log\left[\frac{e^{\sum_{a\in A(y)} o_a}}{1 + \sum_{l\in\mathcal{L}(\mathcal{H}')} e^{\sum_{a'\in A(l)} o_{a'}}}\right] \tag{33}$$

First, we take the derivative of the loss w.r.t ground truth logit $o_y$. We denote the normalization of the first fraction as $Z$.

$$\frac{d\mathcal{L}}{do_y} = -\frac{Z}{e^{\sum_{a\in A(y)} o_a}} \times \frac{Ze^{\sum_{a\in A(y)} o_a} - e^{\sum_{a\in A(y)} o_a}e^{\sum_{a\in A(y)} o_a}}{Z^2} \tag{34}$$

$$= -\frac{Z}{e^{\sum_{a\in A(y)} o_a}} \times \frac{e^{\sum_{a\in A(y)} o_a}}{Z^2} \times \left[Z - e^{\sum_{a\in A(y)} o_a}\right] \tag{35}$$

$$= -\left[1 - \frac{e^{\sum_{a\in A(y)} o_a}}{1 + \sum_{l\in\mathcal{L}(\mathcal{H}')} e^{\sum_{a'\in A(l)} o_{a'}}}\right] \tag{36}$$

The gradient is negative, meaning minimizing the loss function increases the ground truth logit $o_y$. If the value of $o_y$ is low, the gradient magnitude is high. For a high value of all the non-ground truth logits, the gradient magnitude is high. Next, we

take the derivative w.r.t a non-ground truth leaf logit $y'$.

$$\frac{d\mathcal{L}}{do_{y'}} = -\frac{Z}{e^{\sum_{a\in A(y)} o_a}} \times \frac{Z \times 0 - e^{\sum_{a\in A(y)} o_a} e^{\sum_{a'\in A(y')} o_{a'}}}{Z^2} \tag{37}$$

$$= \frac{Z}{e^{\sum_{a\in A(y)} o_a}} \times \frac{e^{\sum_{a\in A(y)} o_a}}{Z^2} \times e^{\sum_{a'\in A(y')} o_{a'}} \tag{38}$$

$$= \frac{e^{\sum_{a'\in A(y')} o_{a'}}}{1 + \sum_{l\in\mathcal{L}(\mathcal{H}')} e^{\sum_{a'\in A(l)} o_{a'}}} \tag{39}$$

The gradient is positive, meaning minimizing the loss function decreases the non-ground truth leaf logit $o'_y$. If the value of $o'_y$ is high, the gradient magnitude is high. Therefore, it guarantees the decrease of all the non-ground truth leaf logits. Next, we take the derivative of the loss w.r.t a non-ground truth non-leaf logit $y'_a$.

$$\frac{d\mathcal{L}}{do_{y'}} = -\frac{Z}{e^{\sum_{a\in A(y)} o_a}} \tag{40}$$

$$\times \frac{Z \times 0 - e^{\sum_{a\in A(y)} o_a}\left[\sum_{y'\in\mathcal{D}(y'_a)\wedge y'\in\mathcal{L}(\mathcal{H}')} e^{\sum_{a'\in A(y')} o_{a'}}\right]}{Z^2} \tag{41}$$

$$= \frac{\sum_{y'\in\mathcal{D}(y'_a)\wedge y'\in\mathcal{L}(\mathcal{H}')} e^{\sum_{a'\in A(y')} o_{a'}}}{Z} \tag{42}$$

The gradient is positive, meaning minimizing the loss function decreases the non-ground truth leaf logit $o_{y'_a}$. If the value of $o_{y'_a}$ is high, the gradient magnitude is also high. With this analysis, all the values of non-ground truth logits $o'_y$ decrease by minimization of the loss function. □

### C.3. Illustration of theorems

In this section, we illustrate the key concept introduced by Theorems 3.1 and 3.4. We use Figure 4 as a representative example. For each of the nodes, we have an associated logit value defined as: Seagull: $o_2$, Sparrow: $o_3$, Openset Bird: $o_4$, Openset Seagull: $o_5$, California Gull: $o_6$, Ivory Gull: $o_7$, Fox Sparrow: $o_8$, Grasshopper Sparrow: $o_9$ and Openset Sparrow: $o_{10}$. Let's take a sample from California Gull, the path probability can be defined as:

$$\text{ph}(\text{California Gull}|x) = \frac{\exp(o_2 + o_6)}{Z_{\text{ph}}} \tag{43}$$

where, $Z_{\text{ph}} = \exp(o_2 + o_5) + \exp(o_2 + o_6) + \exp(o_2 + o_7) + \exp(o_3 + o_5) + \exp(o_3 + o_6) + \exp(o_3 + o_7) + \exp(o_4 + o_5) + \exp(o_4 + o_6) + \exp(o_4 + o_7)$. The loss function that maximizes the path probability $\text{ph}(\text{California Gull}|x)$ is defined as:

$$\mathcal{L}^{\text{ph}} = -\log\left[\frac{\exp(o_2 + o_6)}{Z_{\text{ph}}}\right] \tag{44}$$

In theorem 3.1, there are three types of gradient analysis: (i) gradient for ground truth logit, (ii) gradient for non-ground truth logit, and (iii) gradient for non-ground truth logit with special condition. First, we analyse gradient w.r.t logit of the ground truth class $o_6$:

$$\frac{d\mathcal{L}^{\text{ph}}}{do_6} = -\frac{Z_{\text{ph}}}{\exp(o_2 + o_6)} \times \frac{Z_{\text{ph}}\exp(o_2 + o_6) - \exp(o_2 + o_6)[\exp(o_2 + o_6) + \exp(o_3 + o_6) + \exp(o_4 + o_6)]}{Z_{\text{ph}}^2}$$

$$= -\frac{Z_{\text{ph}} - [\exp(o_2 + o_6) + \exp(o_3 + o_6) + \exp(o_4 + o_6)]}{Z_{\text{ph}}}$$

$$= -\left[1 - \frac{\exp(o_6)[\exp(o_2) + \exp(o_3) + \exp(o_4)]}{[\exp(o_2) + \exp(o_3) + \exp(o_4)][\exp(o_5) + \exp(o_6) + \exp(o_7)]}\right]$$

$$= -\left[1 - \frac{\exp(o_6)}{\exp(o_5) + \exp(o_6) + \exp(o_7)}\right]$$

*Table 8.* Gradient Summary

| Method | Node | Gradient | Remark |
|---|---|---|---|
| Path Prob. | Seagull | $-[1 - \frac{e^{o_2}}{e^{o_2}+e^{o_3}+e^{o_4}}]$ | **Local Optimization**: Magnitude only depends on the ground truth node Seagull and its siblings (Sparrow and Openset Bird). |
| State Prob. | Seagull | $-[1 - \frac{e^{o_1+o_2} \times \{e^{o_5}+e^{o_6}+e^{o_7}\}}{Z_{sp}}]$ | **Global Optimization**: The gradient magnitude depends on all the nodes. |
| Path Prob. | Sparrow | $\frac{e^{o_3}}{e^{o_2}+e^{o_3}+e^{o_4}}$ | **Local Optimization**: Magnitude only depends on the non-ground truth node Sparrow and its siblings (Seagull and Openset Bird). |
| State Prob. | Sparrow | $\frac{e^{o_1+o_3} \times \{e^{o_8}+e^{o_9}+e^{o_{10}}\}}{Z_{sp}}$ | **Global Optimization**: The gradient magnitude depends on all the nodes. |
| Path Prob. | Openset Bird | $\frac{e^{o_4}}{e^{o_2}+e^{o_3}+e^{o_4}}$ | **Local Optimization**: Magnitude only depends on the non-ground truth node Openset Bird and its siblings (Seagull and Sparrow). |
| State Prob. | Openset Bird | $\frac{e^{o_1+o_4}}{Z_{sp}}$ | **Global Optimization**: The gradient magnitude depends on all the nodes. |
| Path Prob. | Openset Seagull | $\frac{e^{o_5}}{e^{o_5}+e^{o_6}+e^{o_7}}$ | **Local Optimization**: Magnitude only depends on the non-ground truth node Openset Seagull and its siblings (California and Ivory Gull). |
| State Prob. | Openset Sparrow | $\frac{e^{o_1+o_2+o_5}}{Z_{sp}}$ | **Global Optimization**: The gradient magnitude depends on all the nodes. |
| Path Prob. | California Gull | $-[1 - \frac{e^{o_6}}{e^{o_5}+e^{o_6}+e^{o_7}}]$ | **Local Optimization**: Magnitude only depends on the ground truth node California Gull and its siblings (Ivory Gull and Openset Seagull). |
| State Prob. | California Gull | $-[1 - \frac{e^{o_1+o_2+o_6}}{Z_{sp}}]$ | **Global Optimization**: The gradient magnitude depends on all the nodes. |
| Path Prob. | Ivory Gull | $\frac{e^{o_7}}{e^{o_5}+e^{o_6}+e^{o_7}}$ | **Local Optimization**: Magnitude only depends on the non-ground truth node Ivory Gull and its siblings ( Openset Seagull and California Gull). |
| State Prob. | Ivory Gull | $\frac{e^{o_1+o_2+o_7}}{Z_{sp}}$ | **Global Optimization**: The gradient magnitude depends on all the nodes. |
| Path Prob. | Fox Sparrow | $0$ | There is no gradient update. |
| State Prob. | Fox Sparrow | $\frac{e^{o_1+o_3+o_8}}{Z_{sp}}$ | **Global Optimization**: The gradient magnitude depends on all the nodes. |
| Path Prob. | Grasshopper Sparrow | $0$ | There is no gradient update. |
| State Prob. | Grasshopper Sparrow | $\frac{e^{o_1+o_3+o_9}}{Z_{sp}}$ | **Global Optimization**: The gradient magnitude depends on all the nodes. |
| Path Prob. | Openset Sparrow | $0$ | There is no gradient update. |
| State Prob. | Openset Sparrow | $\frac{e^{o_1+o_3+o_{10}}}{Z_{sp}}$ | **Global Optimization**: The gradient magnitude depends on all the nodes. |

Here, the gradient is $-ve$, and the gradient magnitude only depends on the siblings of California Gull: Openset Seagull and Ivory Gull. The gradient update increases the value of $o_6$. We can observe that, the increase of the value is dependent only on the local view (the ground truth class and its siblings.)

Next, we analyse gradient w.r.t a non-ground truth class $o_7$.

$$
\begin{aligned}
\frac{d\mathcal{L}^{ph}}{do_7} &= -\frac{Z_{ph}}{\exp(o_2+o_6)} \times \frac{Z_{ph} \times 0 - \exp(o_2+o_6)[\exp(o_2+o_7)+\exp(o_3+o_7)+\exp(o_4+o_7)]}{Z_{ph}^2} \\
&= \frac{\exp(o_2+o_7)+\exp(o_3+o_7)+\exp(o_4+o_7)}{Z_{ph}} \\
&= \frac{\exp(o_7)[\exp(o_2)+\exp(o_3)+\exp(o_4)]}{[\exp(o_2)+\exp(o_3)+\exp(o_4)] \times [\exp(o_5)+\exp(o_6)+\exp(o_7)]} \\
&= \frac{\exp(o_7)}{\exp(o_5)+\exp(o_6)+\exp(o_7)}
\end{aligned}
$$

Here, the gradient is $+ve$, and the gradient magnitude only depends on the ground truth and its siblings (local optimization). Finally, let's analyse gradient w.r.t a non-ground truth class $y' \notin \mathcal{C}(a), \forall a \in \mathcal{A}(y)$ as presented in theorem 3.1. In the representative example of 4, the symbols represent the following for the image of California Gull:

- $y$ = California Gull

- $\mathcal{A}(y)$ = {Bird, Seagull, California Gull}

- $\forall a \in \mathcal{A}(y), \mathcal{C}(a)$ = {Seagull, Sparrow, Openset Bird, Opensest Seagull, California Gull, Ivory Gull}

- Non-ground truth nodes with condition: $y' \notin \mathcal{C}(a), \forall a \in \mathcal{A}(y)$ = { Fox Sparrow, Grasshopper Sparrow, Openset Sparrow }

If we take the gradient w.r.t non-ground truth node with condition, for example $o_8$.

$$\frac{d\mathcal{L}^{\mathrm{ph}}}{do_8} = 0 \tag{45}$$

Since, the gradient value is $0$, there is no update for the non-ground truth logits with condition. It should be noted that, in this particular example, the non-ground truth nodes with condition are from the leaf level in the hierarchy. However, as the depth of hierarchy increases, nodes from non-leaf levels are also included in the non-ground truth nodes with condition.

For theorem 3.4, we have two types of gradient analysis. (i) ground truth logit and (ii) non-ground truth logit. The state probability for California Gull can be defined as:

$$p(\text{California Gull}|x) = \frac{\exp(o_1 + o_2 + o_6)}{Z_{sp}} \tag{46}$$

where $Z_{sp} = 1 + \exp(o_1 + o_4) + \exp(o_1 + o_2 + o_5) + \exp(o_1 + o_2 + o_6) + \exp(o_1 + o_2 + o_7) + \exp(o_1 + o_3 + o_8) + \exp(o_1 + o_3 + o_9) + \exp(o_1 + o_3 + o_{10})$ The corresponding loss function is defined as:

$$\mathcal{L} = -\log\left[\frac{\exp(o_1 + o_2 + o_6)}{Z_{sp}}\right] \tag{47}$$

First, taking gradient for ground truth class, California Gull we have:

$$\frac{d\mathcal{L}}{do_6} = -\frac{Z_{sp}}{\exp(o_1 + o_2 + o_6)} \times \frac{Z_{sp} \times \exp(o_1 + o_2 + o_6) - \exp(o_1 + o_2 + o_6)\exp(o_1 + o_2 + o_6)}{Z_{sp}^2}$$

$$= -\left[1 - \frac{\exp(o_1 + o_2 + o_6)}{Z_{sp}}\right]$$

The gradient is -ve and the magnitude is dependent of all the nodes (global view). Next, taking gradient for non-ground truth class Fox Sparrow:

$$\frac{d\mathcal{L}}{do_8} = -\frac{Z_{sp}}{\exp(o_1 + o_2 + o_6)} \times \frac{Z_{sp} \times 0 - \exp(o_1 + o_2 + o_6)\exp(o_1 + o_3 + o_8)}{Z_{sp}^2}$$

$$= \frac{\exp(o_1 + o_3 + o_8)}{Z_{sp}}$$

We see that all the gradient is +ve and magnitude is dependent on all the nodes. Similarly, there is not any nodes with zero gradient update. The gradient summary of all the nodes for sample of California Gull in the example 4 with the remark is presented in Table 8.

## D. Details of Experiments and Additional Results

In this section, we present more details on the experiments along with some additional results.

### D.1. Details of Datasets

We perform experiments on three real-world hierarchical datasets. The details of each dataset are summarized in Table 9, including (i) the Total number of samples, (ii) the Total number of classes, (iii) Known class split: total number of classes used in the training (iv) Openset class split: total number of classes used in testing as openset classes (v) Number of leaf nodes in the training hierarchy (vi) Number of non-leaf nodes in the training hierarchy and (vii) Maximum depth associated with the training hierarchy.

*Table 9.* Description of real-world hierarchical datasets

| Dataset Name | TinyImagenet | CUB-200-2011 | AWA2 |
|---|---|---|---|
| Total Number of Samples | $120,000$ | $12,000$ | $37,000$ |
| Total Number of Classes | 200 | 200 | 50 |
| Known Class Split | 150 | 150 | 40 |
| Openset Class Split | 50 | 50 | 10 |
| Leaf Nodes in Training Hierarchy | 150 | 150 | 40 |
| Non-Leaf Nodes in Training Hierarchy | 86 | 43 | 21 |
| Max Depth of Training Hierarchy | 12 | 7 | 7 |

*Table 10.* Significance Test

| | CUB | | Tiny Imagenet | | AWA2 | |
|---|---|---|---|---|---|---|
| | **NA@50** | **AUC** | **NA@50** | **AUC** | **NA@50** | **AUC** |
| Baseline | 53.72±0.39 | 45.54±0.08 | 21.42±0.96 | 17.16±0.69 | 48.75±0.23 | 44.86±0.28 |
| State-FGOD | **64.29** $\pm$ 0.17 | **50.79** $\pm$ 0.17 | **24.53** $\pm$ 0.50 | **19.83** $\pm$ 0.39 | **53.24** $\pm$ 0.30 | **47.71** $\pm$ 0.31 |
| p-value | $1.48 \times 10^{-11}$ | $5.77 \times 10^{-12}$ | $2.06 \times 10^{-4}$ | $7.07 \times 10^{-5}$ | $2.87 \times 10^{-12}$ | $3.64 \times 10^{-10}$ |

## D.2. Implementation Details

For the proposed method and baselines, we use ViT-B/16 as the baseline, and prompt as parameter efficient fine-tuning method. We follow the default configuration of prompts, learning rate, and number of epochs as presented by (Jia et al., 2022). For the taxonomy definition, we define a pre-defined numpy file, along with state representation. The state representation defined in the NumPy file is leveraged in defining the loss function 6. The code is written in Python using PyTorch as a library. We conduct all the experiments in NVIDIA A100 with 32GB memory. For the known-novel split, we follow (Pyakurel & Yu, 2024). The conceptual diagram of model training is presented in Figure 6. For the EHND method, we use the value of $\beta_1$ and $\beta_2$ as suggested by the paper (Pyakurel & Yu, 2024). For relabel method, we use the relabing rate of 20%.

## D.3. Significance Test

For the results presented in the main paper, the strongest baseline for CUB is E-HND. Similarly, for TinyImageNet and AWA2, the strongest baseline is Relabel and LOO respectively. In this section, we run our method along with the strongest baseline for 5 different seeds to test whether the improvement of the proposed method is statistically significant. The mean of the performance, the standard deviation of the proposed method, and the strongest baseline, along with the p-value of the t-test is presented in Table 10. The lower value of p-values justifies the statistical significance of the improvement from the proposed method.

## D.4. Additional Results

**Experimental results on Stanford Cars and Cifar100** In this section, we compare the proposed method on additional datasets along with some representative global detectors: LOO, Relabel, and EHND. For Cifar100, we select 40 classes as closed-set classes. We construct the hierarchy of using those 40 classes as leaf classes resulting in 21 non-leaf nodes and a depth of 3. Similarly, for Stanford Cars, we select 166 classes as closed-set classes. We construct the hierarchy using those 166 classes as leaf classes resulting in 10 non-leaf nodes and a depth of 3. The comparison results are presented in Table 11. From the comparison, we see that the performance of the proposed method is superior to the baselines justifying the effectiveness of the proposed methodology.

**Qualitative study.** We start by showing how the path probability mitigates the shortcomings of local detector methods. For an openset sample of Le Conte Sparrow, we use the baseline Hierarchical Classifier with extra nodes with and without path probability in Figure 9(a). The ground truth of the openset sample is Sparrow. We provide probabilities from local detectors in the box of its corresponding child nodes, and path probabilities in blue color at the end of each path. The correct way of inference for local detectors is Bird $\rightarrow$ Passeriform Bird $\rightarrow$ Sparrow. Then, the detector Sparrow should allocate a high openset score so that it would not be further classified to the child node. However, we see that Oscine Bird gets the

*Table 11.* Comparison results on additional datasets

| Method | Stanford Cars | | Cifar100 | |
|---|---|---|---|---|
| | OA@50 | AUC | OA@50 | AUC |
| LOO | 33.15 | 28.31 | 22.71 | 18.38 |
| Relabel | 29.82 | 26.75 | 18.89 | 16.26 |
| EHND | 41.47 | 28.81 | 15.89 | 12.42 |
| State-FGOD | **47.03** | **35.05** | **24.97** | **20.69** |

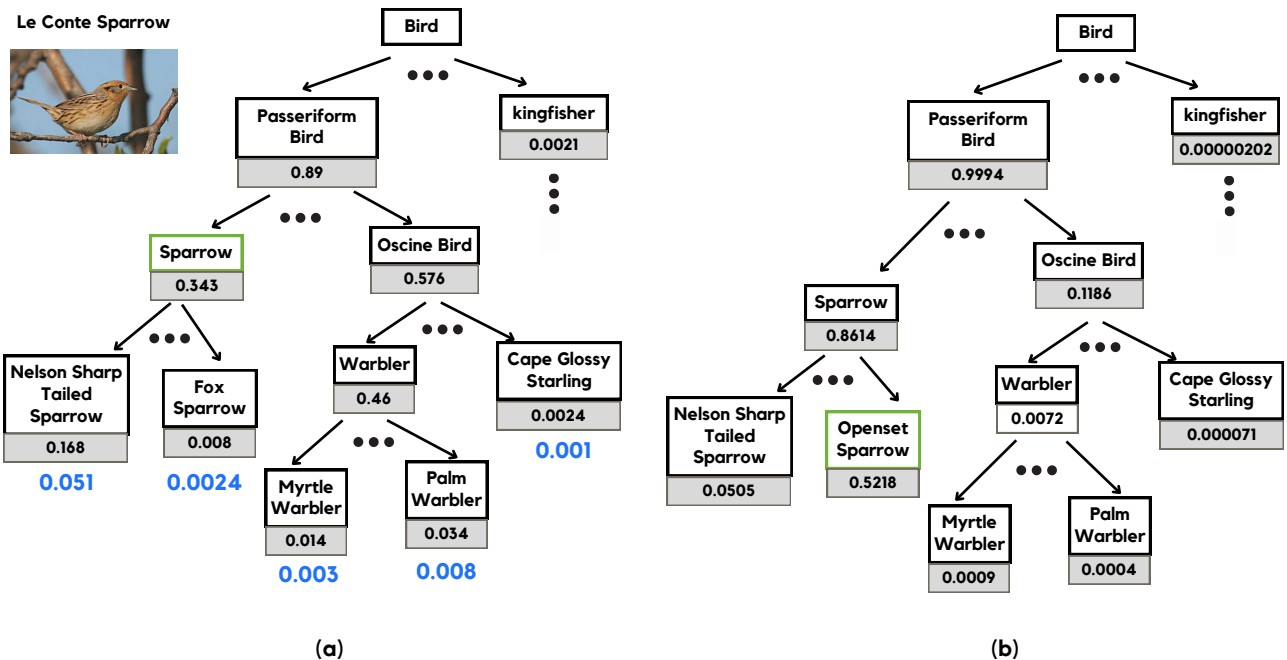

*Figure 9.* Detection result on a sample of Le Conte Sparrow, an openset sample of Sparrow for (a) local detector vs local detector with path probability and (b) State-FGOD

higher probability (0.576) than the correct prediction Sparrow (0.343) and makes the mistake of Fine-Grained Openset Detection. However, if we use path probabilities for inference as provided for all the given paths in Figure 9(a), the path from Bird → Nelson Sharp Tailed Sparrow with path probability (0.051) is selected. Next, the local detector Sparrow has a high openset score of 0.58, and therefore provides the final prediction as Openset Sparrow. In this way, path probability mitigates the error from Passeriform Bird detector.

However, there is still one issue that Sparrow class has a lower probability than Oscine Bird. Although the final prediction is correct, the correct non-leaf nodes still can have a lower probability value than their siblings. If we observe the prediction from the proposed method (State-FGOD) in Figure 9(b), the ground truth Openset Sparrow has the highest probability (0.5218) among all the leaf nodes of the hierarchy. Additionally, non-leaf node Sparrow (0.8614) has the highest probability than its siblings (eg, Oscine Bird with probability of 0.1186). Similar is the case for Passeriform Bird. This probability distribution is possible for the proposed method due to the state formulation as discussed in Proposition 3.3.

For the next openset sample from Chestnut Sided Warbler, we compare the local detector with path probability with the proposed method (State-FGOD) in Figure 10. We observe that one of the child nodes Green Kingfisher has a higher logit value, and high probability, making the path probability higher for the path of Bird → Green Kingfisher highest in Figure 10(a). Because of this, the path probability of one of the child nodes of ground truth node Warbler is not the highest, making Green Kingfisher a wrong final prediction of FGOD. However, in Figure 10(b), we observe that the proposed method assigns the highest logit to Openset Warbler, making a correct FGOD prediction.

**Computational Cost Analysis:** We conducted a computational cost analysis. We define the input dimension of the classifier head as $D$, the total number of leaf nodes in the class hierarchy as $N$, and the depth of the hierarchy as $h$. For ViT

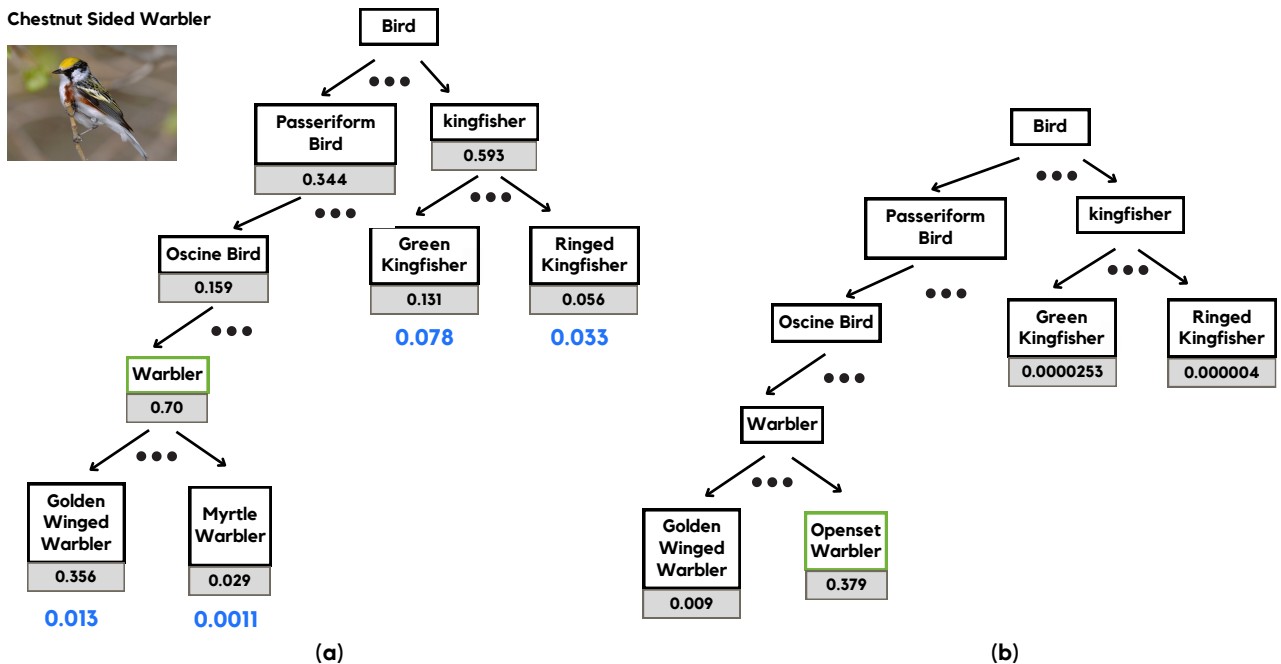

*Figure 10.* Detection result on a sample of Chestnut Sided Warbler, an openset sample of Warbler for (a) local detector with path probability vs (b) State-FGOD

and ResNet backbone, the value of $D$ is $768$ and $2048$, respectively. For the TinyImagenet, CUB, and AWA2, the value of $N$ is $236$, $193$ and $61$, respectively. The corresponding values of $h$ for these datasets are $12$, $7$, and $7$, respectively. So, we note that $D > N \gg h$.

For local detectors, we assume the output dimension of each classifier to be $n$ on average, which is no more than $5$ in our datasets. So, we have $n \ll N$. The time complexity of the logit calculation is $O(D \times n \times h)$. The top-down inference for receiving the final prediction takes $O(n \times h)$. The total complexity is $O(D \times n \times h) + O(n \times h)$, and overall complexity is $O(D \times n \times h)$. For global detectors, the classifier output dimension is $N$, resulting in a logit calculation complexity of $O(D \times N)$. The prediction complexity is $O(N)$. Hence, we obtain a total complexity of $O(D \times N) + O(N)$ and an overall complexity of $O(D \times N)$. For the proposed method, we need to aggregate the probabilities by utilizing the global states, which is an extra step compared to global detectors. This is obtained using matrix multiplication of logits and global states, adding the complexity of $O(N^2)$. Thus, the total complexity is $O(D \times N) + O(N) + O(N^2)$. For the datasets we used, we have $D > N$, resulting in an overall complexity of $O(D \times N)$, which is the same as the global detectors.

We use the ViT backbone to calculate the inference time (in milliseconds for one test sample) for the proposed method and representative baselines. The analysis is presented in Table 12. The total inference includes the time of both the logits calculation and the final prediction. Since the number of leaf classes follows the order $N_{Tiny} > N_{CUB} > N_{AWA2}$, the corresponding inference times (both total and logit calculation) adheres to the same trend $\text{Time}_{Tiny} > \text{Time}_{CUB} > \text{Time}_{AWA2}$. Among all the datasets, local detectors require less time than global detectors, which is due to smaller output dimension $n < N$ and top-down inference $h < N$. Finally, the overall complexity analysis revealed that the logits calculation has the dominant time complexity. This is reflected in a substantial portion of the total inference time occupied by logits calculation. In comparison to other global detector methods, the proposed method requires slightly higher inference time, which is due to extra step of calculating global state probabilities. Overall, the inference time of all the methods is fairly fast, making them usable in a practical setting.

## E. Limitations, and Future Work

In this work, we evaluate the proposed approach using various benchmark datasets to validate the superior performance of the proposed method. We acknowledge that the real openset dataset is unbounded for a given training hierarchy. To capture the nature of such data, we utilize the samples from the closed-set classes to train open-set classes as well. It

Table 12. Inference and Logits Calculation Time (in milliseconds)

| Method | Metric | TinyImagenet | CUB | AWA2 |
|---|---|---|---|---|
| TD | Total inference | 0.0009189 | 0.0007314 | 0.0002797 |
| | Logits calculation | 0.0007530 | 0.0005900 | 0.0001950 |
| MSP | Total inference | 0.0009213 | 0.0007297 | 0.0002807 |
| | Logits calculation | 0.0007530 | 0.0005900 | 0.0001950 |
| HC-EN | Total inference | 0.0009225 | 0.0007464 | 0.0002794 |
| | Logits calculation | 0.0007530 | 0.0005900 | 0.0001950 |
| LOO | Total inference | 0.3191200 | 0.2956500 | 0.1330400 |
| | Logits calculation | 0.3138200 | 0.2930120 | 0.1304500 |
| E-HND | Total inference | 0.3186700 | 0.2959900 | 0.1341200 |
| | Logits calculation | 0.3138200 | 0.2930120 | 0.1304500 |
| STATE-FGOD | Total inference | 0.3204700 | 0.2984500 | 0.1365200 |
| | Logits calculation | 0.3138200 | 0.2930120 | 0.1304500 |

would be interesting to generate meaningful openset samples and use them to augment training data to further improve the performance of FGOD. We will investigate the data augmentation to aid fine-grained openset dataset as a future work.

## F. Source Code

The source code is provided in the repository: https://github.com/ritmininglab/STATE-FGOD

