# OpenReview forum: "Learning State-Based Node Representations from a Class Hierarchy for Fine-Grained Open-Set Detection"
_ICML.cc/2025/Conference — ICML 2025 poster_

### Official Review · Reviewer_YgaW · 2025-03-08

**Overall Recommendation:** 4

**Summary:**

This paper introduces a structured state representation for fine-grained open-set detection (FGOD). Prior approaches fall into two categories: local and global detectors. Local detectors iteratively multiply probabilities along a tree path but suffer from error accumulation. Global detectors flatten probabilities and predict all at once but ignore tree structures. The proposed State-FGOD addresses these issues by combining both approaches: 1) computing node logits for all tree paths, then 2) normalizing them to obtain the global probability. The paper provides theoretical justifications for previous limitations and demonstrates State-FGOD achieving state-of-the-art performance on FGOD benchmarks, supported by hierarchical performance measures.

**Claims And Evidence:**

All claims are well supported.

**Essential References Not Discussed:**

The related works are generally well-surveyed, including recent studies. The only missing one I noticed is InfoSieve [1], which addresses fine-grained category discovery by unsupervisedly learning codes (or states) for the class tree. However, InfoSieve targets a related but different problem and does not leverage known class structures, distinguishing it from this paper. Therefore, citing it is not necessary.

[1] Learn to Categorize or Categorize to Learn? Self-Coding for Generalized Category Discovery. NeurIPS 2023.

**Experimental Designs Or Analyses:**

Well conducted.

**Methods And Evaluation Criteria:**

Intuitive and clearly explained.

**Other Comments Or Suggestions:**

N/A

**Other Strengths And Weaknesses:**

I believe the paper addresses an important yet underrecognized problem, and the proposed state representation seems to be an effective approach to solving it. Therefore, I think the paper is worth accepting as a reference for future research.

---
One downside is its computational cost, as it aggregates computations over all leaf nodes, making it more expensive than local and global detectors. Discussing inference complexity and including a table comparing computation times across different approaches would be useful.

With $N$ leaf nodes in a well-balanced class tree of depth $\log N$, the computational complexity is $O(\log N)$ for local detectors, $O(N)$ for global detectors, and $O(N \log N)$ for this approach, which remains manageable for large-scale classes. In the worst case, where depth is $O(N)$, complexity could reach $O(N^2)$, but most practical cases likely involve a reasonably balanced tree.

**Questions For Authors:**

N/A

**Relation To Broader Scientific Literature:**

FGOD is a practical yet underexplored problem in the ML community. This paper could have a broader scientific impact, such as in fine-grained bird species recognition.

**Theoretical Claims:**

It seems correct.

---

> ### Author Rebuttal · Authors · 2025-03-30
>
> Thank you for taking the time to review our paper. We appreciate your constructive comments. Our main responses are summarized below.
>
> **Q1: Computational Cost**
>
>  Thank you for the comment. We conducted a computational cost analysis. We define the input dimension of the classifier head as $D$, the total number of leaf nodes in the class hierarchy as $N$, and the depth of the hierarchy as $h$. For ViT and ResNet backbone, the value of $D$ is $768$ and $2048$, respectively.  For the TinyImagenet, CUB, and AWA2, the value of $N$ is $236$, $193$ and $61$, respectively. The corresponding values of $h$ for these datasets are $12$, $7$, and $7$, respectively. So, we note that $D>N\gg h$.
>
> For local detectors, we assume the output dimension of each classifier to be $n$ on average, which is no more than 5 in our datasets. So, we have $n\ll N$. The time complexity of the logit calculation is $O(D\times n\times h)$. The top-down inference for receiving the final prediction takes $O(n \times h)$. The total complexity is $O(D\times n\times h)$ + $O(n \times h)$, and overall complexity is $O(D\times n\times h)$. For global detectors, the classifier output dimension is $N$, resulting in a logit calculation complexity of $O(D\times N)$. The prediction complexity is $O(N)$. Hence, we obtain a total complexity of $O(D\times N) + O(N)$ and an overall complexity of  $O(D\times N)$. For the proposed method, we need to aggregate the probabilities by utilizing the global states, which is an extra step compared to global detectors. This is obtained using matrix multiplication of logits and global states, adding the complexity of $O(N^2)$. Thus, the total complexity is  $O(D\times N) + O(N)+ O(N^2)$. For the datasets we used, we have $D> N$, resulting in an overall complexity of $O(D\times N)$, which is the same as the global detectors.
>
> We use the ViT backbone to calculate the inference time (in milliseconds for one test sample) for the proposed method and representative baselines. The total inference includes time  of both the logits calculation and final prediction.  Since the number of leaf classes follows the order $N_{Tiny} > N_{CUB} > N_{AWA2}$,  the corresponding inference times (both total and logit calculation) adheres to the same trend $Time_{Tiny} > Time_{CUB} > Time_{AWA2}$. Among all the datasets, local detectors require less time than global detectors, which is due to smaller output dimension $n < N$ and top-down inference $h < N$.  Finally, the overall complexity analysis revealed that the logits calculation has the dominant time complexity. This is reflected in a substantial portion of the total inference time occupied by logits calculation. In comparison to other global detector methods, the proposed method requires slightly higher inference time, which is due to extra step of calculating global state probabilities. Overall, the inference time of all the methods is fairly fast, making them usable in a practical setting.
>
> | Method     | Time  | TinyImagenet  | CUB    | AWA2        |
> |------------|----------------------|---------------|-------------|-------------|
> | TD      | Total inference      | 0.0009189     | 0.0007314   | 0.0002797   |
> |            | Logits calculation   | 0.0007530     | 0.0005900   | 0.0001950   |
> | MSP        | Total inference      | 0.0009213     | 0.0007297   | 0.0002807   |
> |            | Logits calculation   | 0.0007530     | 0.0005900   | 0.0001950   |
> | HC-EN      | Total inference      | 0.0009225     | 0.0007464   | 0.0002794   |
> |            | Logits calculation   | 0.0007530     | 0.0005900   | 0.0001950   |
> | LOO        | Total inference      | 0.3191200     | 0.2956500   | 0.1330400   |
> |            | Logits calculation   | 0.3138200     | 0.2930120   | 0.1304500   |
> | E-HND      | Total inference      | 0.3186700     | 0.2959900   | 0.1341200   |
> |            | Logits calculation   | 0.3138200     | 0.2930120   | 0.1304500   |
> | STATE-FGOD | Total inference      | 0.3204700     | 0.2984500   | 0.1365200   |
> |            | Logits calculation   | 0.3138200     | 0.2930120   | 0.1304500   |
>
> **Q2: Reference not discussed.**
>
> Thank you for suggesting the reference of Generalized Category Discovery (GCD) [1]. GCD and FGOD both target a related problem of assigning openset samples to a specific category. FGOD leverages an existing hierarchical structure and defines openset class for every non-leaf class in the hierarchy so that openset test samples can be assigned to those openset classes. GCD, on the other hand, does not leverage existing hierarchy but learns to categorize the classes into a hierarchical structure. When a hierarchy is not present for the dataset, GCD approaches can be applied to obtain the hierarchy before the FGOD methods are applied. In that sense, GCD and HND are complementary to each other. We will include a discussion in the revised papper.
>
>
>
> [1] Learn to Categorize or Categorize to Learn? Self-Coding for Generalized Category Discovery. NeurIPS 2023.

---

### Official Review · Reviewer_hS4h · 2025-03-13

**Overall Recommendation:** 4

**Summary:**

This paper handles fine-grained open-set detection, which features a hierarchical structure of semantic classes. The authors discussed the limitations of existing local and global detectors, with theoretical analyses unveiling the suboptimality of these detectors. Then a state-based approach is proposed by the authors to effectively capture the dependencies among the nodes in the hierarchy. A formal analysis is provided to validate the optimal training process, which is further supported by significant performance gains in the experiments.

**Claims And Evidence:**

Yes. The authors have provided detailed theoretical analysis and experiment evidences.

**Essential References Not Discussed:**

To my limited knowledge, no related works are missing.

**Experimental Designs Or Analyses:**

The authors have clearly clarified the experiment settings including datasets, baseline setup, and evaluation metrics. The effectiveness of the proposed method is supported by the performance gains over baseline methods. And the ablations on sub-modules / designs are also provided.

**Methods And Evaluation Criteria:**

Yes

**Other Comments Or Suggestions:**

No other comments.

**Other Strengths And Weaknesses:**

The paper is well written with clear motivation and smooth logic flow.

**Questions For Authors:**

Just out of curiosity, can the authors explain the difference between open-set detection, near-OOD detection and OOD detection?

**Relation To Broader Scientific Literature:**

The findings in this paper on hierarchical openset detection are related to hierarchical classification that handles closedset semantic categories.

**Theoretical Claims:**

The theoretical analysis of the local/global detector and the proposed state-based approach was checked, and no issues were found.

---

> ### Author Rebuttal · Authors · 2025-03-30
>
> Thank you for taking the time to review our paper. We appreciate your constructive comments. Our main responses are summarized below.
>
> **Q1: Difference between open-set detection, near-OOD detection, and OOD detection?**
>
> The goal of open set detection is to identify test samples that are semantically different from the closed-set classes used in the training. Some studies also refer to open-set classes as Out-Of-Distribution (OOD) and closed-set classes as in-distribution. In that sense, the goal of OOD detection is the same as open set detection. That being said, Out-of-distribution is also a more general term used to refer to any test distribution that does not match the training distribution. To this end, the definition of OOD is not limited to open set classes but may also refer to label shift, subpopulation shift, or domain shift as well.
>
> Based on how similar open-set classes are with the closed-set one, open-set detection can be categorized into near-OOD and far-OOD detection. For near-OOD detection, although open-set classes are semantically different classes, they share similarities with closed-set classes.  As an example, the closed-set classes include Husky and Golden Retriever and openset classes may include other types of dogs, wuch as German Shepherd and Bulldog. For far-OOD detection, open-set classes are much more distinct from the closed-set ones. For example, the closed-set classes may include cars and airplanes while the open-set ones may cover different animals.

---

### Official Review · Reviewer_JNfG · 2025-03-17

**Overall Recommendation:** 3

**Summary:**

This paper addresses the challenge of Fine-Grained Open-Set Detection (FGOD) by leveraging hierarchical structures among known classes. It introduces a state-based node representation that systematically captures fine-grained dependencies between different hierarchical levels. Unlike traditional local or global detectors, this approach reduces error accumulation and ensures optimal training through a mathematically justified method.

**Claims And Evidence:**

Yes

**Essential References Not Discussed:**

No

**Experimental Designs Or Analyses:**

Yes, the main results(Table 1)， the experiments is suitable

**Methods And Evaluation Criteria:**

Yes

**Other Comments Or Suggestions:**

In fact, I am not familiar with this field, so I am unable to provide an accurate assessment of the contributions of this paper. Please, AC, prioritize the opinions of other reviewers.

**Other Strengths And Weaknesses:**

Strengths:
1. The author analyzes the shortcomings of the baseline through intuitive explanations and rigorous proofs, thereby introducing the method they propose. This style of writing aligns well with scientific research practices and is persuasive.
2 The proposed method exhibits relatively superior performance.

Weakness:
The writing could still be optimized.

**Questions For Authors:**

No

**Relation To Broader Scientific Literature:**

Open-Set Recognition (OSR): Traditional OSR approaches focus on detecting whether a test sample is from a known or unknown category. Earlier works like OpenMax  estimated open-set likelihood by recalibrating softmax scores. Other methods, such as Maximum Softmax Probability (MSP) and Evidential Learning , improved uncertainty quantification for OSR.
Hierarchical Classification: Standard hierarchical classification methodsstructured predictions according to taxonomy trees but were limited to closed-set settings.

**Theoretical Claims:**

For theorem 3.1: Simply maximizing the probability of the path leading to the ground-truth leaf node does not guarantee a decrease in logits of incorrect nodes that are not children of the nodes in the path. The proof assumes that only local parent-child constraints affect optimization, but does not explicitly discuss how information from other hierarchical paths (e.g., sibling relationships) might indirectly contribute to improved training.

---

> ### Author Rebuttal · Authors · 2025-03-30
>
> Thank you for taking the time to review our paper. We appreciate your constructive comments. Our main responses are summarized below.
>
>
> **Q1: Sibling relationship for theorem 3.1**
>
> Path probability based optimization (Theorem 3.1) indeed considers sibling relationships. As we mentioned in the proof sketch, gradient update is dependent on the logits of the children of its parent only. For a class, children of its parent refers to its siblings. So, the sibling relationship contributes to the training. However, it does not guarantee a decrease of logits of other incorrect nodes that are not children of the nodes in the path. The failure to consider other incorrect nodes is the reason for sub-optimality.
>
> In addition, when we provide details of the proof in the appendix, we show that the gradient of the ground truth label (Equation 24 and lines 767-769) is dependent on $o_{sb}$, which refers to the logit of sibling $sb$.
>
> **Q2:  The writing could be optimized.**
>
> We will carefully incorporate all the reviewers' suggestions to improve the presentation of the revised paper.

---

### Official Review · Reviewer_1R2j · 2025-03-18

**Overall Recommendation:** 3

**Summary:**

This paper considers fine-grained open-set detection. The proposed approach utilizes hierarchical relationships to determine if a sample does not belong to any of the known classes, but still might be related to them, by e.g. sharing a superclass with some of them. The paper proposes to leverage the hierarchical structure of labels to learn state-based node representations.
The hierarchy is represented as a graph where the nodes represent possible classes. The set of possible classes is extended with open-set classes - each non-leaf node gets an extra child node.  One legal state consists of nodes that may be active at the same time. The model outputs a logit for each node. The probability of the state is the product of exponent of logits of the state's active nodes normalized with probabilities of all possible legal states. The model is trained to maximize the probability of the correct state.

# Score after rebuttal
I have increased my rating after the rebuttal as the authors have addressed my main concerns regarding the training.

**Claims And Evidence:**

The paper claims that the proposed probability formulation considers all of the nodes during training, as opposed to conventional approaches that train independent classifiers along the hierarchy. This is shown through two proofs that seem correct. Good experimental results further support the proposed method.

**Essential References Not Discussed:**

[a] Li et al. Hierarchical semantic segmentation, CVPR 2022, this paper also enforces coherent probabilities along hierarchies

**Experimental Designs Or Analyses:**

The experimental setup follows previous work and seems sound.

**Methods And Evaluation Criteria:**

The proposed approach is evaluated on three fine-grained datasets and follows previous work.

**Other Comments Or Suggestions:**

No comments

**Other Strengths And Weaknesses:**

The paper is somewaht difficult to follow at certain points. A lot of space is dedicated to describing local detection methods leaving some parts of the proposed method unclear, especially with regards to training. See questions

**Questions For Authors:**

I am still unclear about exact training of the method.
Q1) Eq 46 seems to suggest that p(y|x) is equal to p(s|x) and that there is a single model that produces those probabilities. Eq 6 on the other hand contains different parameters that depend on dynamic hierarchies, that are not explained anywhere.
Q2)  Also, p(y|x) is not defined previously.
Q3) It is also not clear to me how the "open-set" nodes are trained exactly. If only the closed set classes are used during training, I would expect those nodes to always have a gradient which lowers them and that they would "die out" given enough training. What is the signal that would increase the logits for those nodes? Algorith 2 mentions a bieas that is added during inference? How is that bias determined?

**Relation To Broader Scientific Literature:**

The method improves on previous work on fine-grained open-set detection.

**Theoretical Claims:**

I went over the proofs in the appendix and they seemed correct.

---

> ### Author Rebuttal · Authors · 2025-03-30
>
> Thank you for taking the time to review our paper. We appreciate your constructive comments. Our main responses are summarized below.
>
> **Q1: Reference not discussed**
>
> Thank you for suggesting this relevant work.  We would like to clarify that the primary focus of this work is to perform hierarchical classification at the pixel level. For example, for a pixel that is part of a Fox Sparrow, the goal is to identify it as Fox Sparrow, Sparrow and Bird as present in the class hierarchy: `Bird → Sparrow → Fox Sparrow`. Different from ours, this work does not consider the open-world setting. In contrast, our goal is to tackle the more challenging openset detection problem by leveraging a known class hierarchy. For the same training hierarchy as mentioned above, we aim to identify a  Le Conte Sparrow as an openset sample, since it does not belong to any known classes in the hierarchy. Our approach first detects the sample as openset, and then identifies the closest parent class (i.e., Sparrow).  We will include a discussion of this work in the related work section.
>
> **Q2: Clarification of the training process**
>
> The design and the training process of the proposed State-FGOD model are presented in Section 3.3. Before describing the training process (right above Theorem 3.4), all the key technical concepts are presented in detail in earlier parts of this section. The overall training loss is shown in equation (6). During the training process, we remove one node $v_m$ at a time. This creates a dynamic hierarchy $\mathcal{H}' \setminus v_m$ (lines 315-316 in the paper and more examples are shown here: https://anonymous.4open.science/r/FGOD_training-37D9/README.md). When $v_m$ is removed, all the samples that belong to $v_m$ will be treated as openset samples with respect to $\mathcal{H}' \setminus v_m$. The goal is to learn a more compact representation of $\mathcal{H}' \setminus v_m$ that can help to detect openset samples that fall outside of this hierarchy.  For a training input $x$ with ground truth $y$, if $v_m = y$ or $v_m \in \mathcal{A}(y)$, then the ground truth label is changed to the openset class as $y = \mathcal{O}(\mathcal{P}(v_m))$. Otherwise the ground truth label is not changed, and the sample belongs to closed-set classes.  For each removed node $v_m$ and training sample $(x,y)$, the training loss maximizes ground truth node probability $p(y|x, \theta^{\mathcal{H'}\setminus v_m})$.
>
> **Q3: Calculation of probability $p(y|x)$ in dynamic hierarchies**
>
> We present the calculation of state probability of California Gull in equation (46). This is equivalent to probability $ p(y|x, \theta^{\mathcal{H'}})$, where sample $x$ belongs to the California Gull class.  However, in equation (6), it contains different parameters that depend on dynamic hierarchies, which are defined in lines 315-316 in the paper and further clarified in our response to Q2.
> As reviewer mentioned, we obtain logits from a single model. We aggregate these logits to obtain unnormalized state probabilities as given by equation (2). The calculation of probability in dynamic hierarchy $p(y|x, \theta^{\mathcal{H'}\setminus v_m})$ leverages the same unnormalized state probabilities and only differs in normalization factor $Z$.  To obtain $Z$ when $v_m$ is removed, we only sum unnormalized probabilities of those states, which remain in the dynamic hierarchy $\mathcal{H'}\setminus v_m$.
>
> **Q4: Definition of $p(y|x)$**
>
>  For a sample with ground truth $y$, $p(y|x, \theta^{\mathcal{H'}})$ is the node probability (lines 310-311 in paper) in the hierarchy $\mathcal{H'}$. When node $v_m$ is removed, $p(y|x, \theta^{\mathcal{H'}\setminus v_m})$ is defined to represent node probability in the corresponding dynamic hierarchy. Node probability is the sum of all the state probabilities where node $v_m$ is active, as explained in lines 289-290. For any ground truth node $y$, only one state is active, making the node probability equal to state probability corresponding to the node $y$.
>
> **Q5: how the "open-set" nodes are trained**
>
> As explained in our response to Q2, we remove one node $v_m$ at a time during model training. All the samples that belong to the removed node will play the role as openset samples with respect to $\mathcal{H}' \setminus v_m$.
>
> **Q6: Role of bias**
>
> Thank you for raising this point. The role of bias is explained in lines 374-375. There is a trade-off between closed and open set performances. To capture the trade-off in evaluation, we define a fixed set of biases $\mathbf{b}$ in [-20, 20].  For each $b\in\mathbf{b}$, we add this term to the logits of open-et nodes. When the value is positive, it favors openset nodes in prediction, and when the value is negative, it favors closed-set nodes. For each $b$, we obtain a pair of closed-set and openset accuracy. We plot them to obtain a curve and use Area Under Curve as an overall evaluation metric.

---

> > ### Comment · Reviewer_1R2j · 2025-04-03
> >
> > I unfortunately still do not understand the training completely.
> >
> > Equation 6 reads to me as training an ensemble of |V| different models with parameters $\theta^{\mathcal{H' \setminus v_m}}$? However for inference, we need a single model with parameters $\theta^{\mathcal{H'}}$. I do not quite understand how we can get $\theta^{\mathcal{H'}}$ from an ensemble $\theta^{\mathcal{H' \setminus v_m}}$.

---

> > > ### Author Response · Authors · 2025-04-03
> > >
> > > Thank you for taking the time to read our rebuttal. We also appreciate your follow-up question regarding the training process, which we understand may have been caused by confusion in the notation.
> > >
> > > To clarify: we train **a single model** parameterized by $\theta$. Each time a node $v_m$ is removed, it creates a new dynamic hierarchy, denoted as $\mathcal{H}' \setminus v_m$. The same model $\theta$ is then trained to detect openset samples that lie outside this modified hierarchy. Since the openset samples can occur at different levels or positions within the class hierarchy, this process allows the model to learn to detect diverse types of openset samples, depending on their specific location within the hierarchy. It is designed to address the unique requirement for the fine-grained openset detection as described in the paper.
> > >
> > > So, the notation $\theta^{\mathcal{H}' \setminus v_m}$ refers to the same model $\theta$ trained using the data corresponds to the dynamic $\mathcal{H}' \setminus v_m$, i.e., when node $v_m$ is removed from $\mathcal{H}'$. To make this clearer and reduce potential confusion, we propose changing the loss function notation from $p(y |x, \theta^{\mathcal{H}' \setminus v_m})$ to $p(y |x, \theta, \mathcal{H}' \setminus v_m)$ in order to separate the model parameters from the hierarchy-dependent data.
> > >
> > > We hope this explanation resolves the confusion. If all of your concerns have been adequately addressed, we would sincerely appreciate it if you could consider updating your rating accordingly.

---

### Decision · Program_Chairs · 2025-05-01

**Decision:**

Accept (poster)

**Comment:**

The paper was reviewed by four experts in the field and finally received all positive scores: Weak accept, Weak accept, Accept and Accept.
The major concerns of the reviewers are:
1.	some details about the training of the method,
2.	the efficiency of the proposed method.
The authors address the above concerns during the discussion period. Hence, I make the decision to accept the paper.